# Revision of the Subgenus *Ochthomantis* Frogs from Madagascar (Amphibia: Mantellidae) with the Description of Four Species and Resurrection of *Mantidactylus catalai* and *M. poissoni* [note 1]

**DOI:** 10.3390/ani13172800

**Published:** 2023-09-03

**Authors:** Nirhy H. C. Rabibisoa, Rachel S. Welt, Christopher J. Raxworthy

**Affiliations:** 1Sciences de la Vie et de l’Environnement, Faculté des Sciences, de Technologies et de l’Environnement, Université de Mahajanga, Campus Universitaire d’Ambondrona, BP 652, Mahajanga 401, Madagascar; 2Department of Herpetology, American Museum of Natural History, Central Park West at 79th Street, New York, NY 10024-5192, USA

**Keywords:** *Ochthomantis*, *Mantidactylus*, cryptic species, morphology, 16S, Madagascar

## Abstract

**Simple Summary:**

The genus *Mantidactylus* spp. is one of the exceptionally diverse amphibian clades from Madagascar. Currently, 57 species in 6 subgenera are recognized. One subgenus, *Ochthomantis*, is the focus of the present study. Here, we revise this taxonomic group to recognize the presence of cryptic species through an assessment of morphological and, where available, molecular (16S mitochondrial gene) variation of 637 sexed adult specimens. Our results show that *Ochthomantis* contains eleven species, including the resurrected *Mantidactylus catalai* and *M. poissoni*, and four newly described species. We cannot confirm that *M. majori* should be considered as part of the subgenus *Ochthomantis*, so we do not include it in our descriptions. Following our taxonomic revision, we also present a practical simple key to identify all of the species belonging to this subgenus.

**Abstract:**

The subgenus *Ochthomantis* is an obligate forest and stream-dwelling group of mantellid frogs, endemic to Madagascar, with six species currently recognized. However, this group suffers from ongoing taxonomic confusion due to low numbers of examined specimens, and failure to consider morphological variation from development and sexual dimorphism. Here, we examined the morphology of 637 sexed adult specimens collected by us in the field and from other museum collections. We also sequenced a DNA fragment of the 16S mtDNA gene for each lineage to determine congruence between morphological and molecular data sets and to help delimit species. Our results demonstrate that the subgenus *Ochthomantis* includes eleven valid species: five already recognized, *M. catalai* and *M. poissoni* that we resurrect from synonymy, and four new species which we describe for the first time here. In some analyses, *Mantidactylus majori* groups with other *Mantidactylus* subgenera, so we do not consider it a member of the subgenus *Ochthomantis* in this study. All species have restricted distributions and elevational ranges in the humid forests of Madagascar. This study demonstrates the utility of assessing cryptic species using both diagnostic morphological characters and molecular data. The discovery of this new cryptic biodiversity, and the taxonomic revision herein, will likely require conservation activities for those species with the most restricted distributions.

## 1. Introduction

Within the diverse radiation of *Mantidactylus* frogs (Anura: Mantellidae) in Madagascar, 57 species have been described [1] across 6 subgenera [2,3,4]. One of these is the subgenus *Ochthomantis* which currently contains six valid species: *Mantidactylus femoralis*, *M. mocquardi*, *M. ambreensis*, *M. ambony, M. majori*, and *M. zolitschka* [5,6]. *Ochthomantis* is characterized by the other *Mantidactylus* subgenera by the combination of the following unique set of characteristics: no webbing between fingers, webbing between toes, large tympanum (more than half the eye diameter), sexual dimorphism in size of tympanum (smaller in females) and body size (larger in females), presence of an inguinal pale streak marking, and living close to rivers [2,6,7]. The *Ochthomantis* group has been recorded from a broad range of elevations from sea level to 2600 m and occurs in most regions of Madagascar except the dry south and west.

The first species description for this group was made by Boulenger [8] for *Mantidacylus femoralis* and this author later also described the genus *Mantidactylus* [9]. Thereafter, additional species were described and some of them later synonymized [4,6]. The subgenus *Ochthomantis* was described by [7,10], with the most recently described species being *M. zolitschka* [10] and *M. ambony* [5], and a review was conducted by [2,3] who proposed a new taxonomy of Malagasy mantellines based on molecular results. The most recent molecular study tentatively suggests that *Mantidactylus argenteus* might also belong to the *Ochthomantis* clade [11].

Many specimens of the *M. femoralis* complex have been collected since 1989 by different researchers such as Ronald Nussbaum’s team (University of Michigan, Christopher Raxworthy’s team (American Museum of Natural History) and Nirhy Rabibisoa and others at the University of Antananarivo. In these collections, it has become clear that *M. femoralis* and other *Ochthomantis* species include distinct morphospecies that are strong candidates for unrecognized additional species. Similarly, other authors have indicated that the group likely includes unrecognized species based on molecular divergence for the 16S rRNA locus and morphological differences [10,11]. A past review of mantellid species using morphological characters has demonstrated that undescribed species likely exist [12]. Therefore, a study utilizing both morphological characters and molecular data is important to resolve the classification of the subgenus *Ochthomantis*. Thus, this study aims to address this issue of cryptic species and to validate candidate species by undertaking a morphological analysis of 637 adult *Ochthomantis* specimens combined with an analysis of genetic data and a detailed historical taxonomic review of all species in the group.

Based on the findings from our study, we here resurrect two species that were previously considered junior synonyms: *M. catalai* and *M. poissoni*; and describe four new species. According to these results, we conclude that the subgenus *Ochthomantis* contains at least 11 species.

## 2. Materials and Methods

Field surveys were carried out during the rainy season (January–May) at numerous sites in Madagascar, except in the Moramanga Region where fieldwork took place during the cooler dry season (July–September) and continued during the rainy season (November–December) to provide additional data on ecology, behavior, and reproduction. All specimens of *Ochthomantis* were collected between 1989 and 2010 within rainforest and deciduous forests in Northwestern and Northern Madagascar. Frogs and tadpoles were collected during both the day and the night using headlamps by searching on the ground and vegetation up to 2 m in height, along riverbanks, in rivers streams, ponds, and lakes. The following information was recorded at the time of capture for each individual: date, time, temperature, elevation (using GPS and altimeter), and microhabitat characteristics. Photographs of representative specimens were taken to record coloration in life. Vouchers specimens collected by the authors were euthanized using chlorotone, fixed using 10% formalin, and later stored in 70% ethanol. Liver and/or thigh muscle was removed from representative specimens before fixation and frozen into liquid nitrogen or preserved in alcohol for molecular analysis. The field tag series abbreviations used in this study are: APR, Achille P. Raselimanana; FAZC and FN, Franco Andreone; J.B. and MRJ, J. B. Ramanamanjato; JR, Jeannot Razafimanantsoa; LLS and LV, Olivier S. Ramilison; MA, Mirana Anjeriniaina; NR, Nirhy H. Rabibisoa; RD and SMG, Domoina Rakotomalala; RJS, Jasmin Randianirina; RAN, Ronald A. Nussbaum; RAX, Christopher J. Raxworthy. A total of 637 specimens were examined in detail, collected by us, and housed at the following collections: AMNH (American Museum of Natural History, Department of Herpetology), BM (The British Museum, London), MNHN (Muséum National d’Histoire Naturelle de Paris); UADBA (University of Antananarivo Department of Animal Biology), PBZT (Parc Botanique et Zoologique de Tsimbazaza Antananarivo), and UMMZ (University of Michigan, Museum of Zoology). Museum collections are abbreviated using standard institutional abbreviations as listed in [13], with the addition of UADBA and PBZT.

Sex and maturity were determined based on examination of the gonads and development of the femoral glands. Juveniles (J) were not included in the morphological studies reported here and males (M) and females (F) were analyzed separately due to the sexual dimorphism in this group. Morphological measurements were made by N.R. using calipers to 0.05 mm precision and a binocular microscope. Each measurement was made as described by [14]. The definition of each character measured followed [15]. Abbreviations used for morphological measurements are given in Appendix B, Table A1, and Figure A1. Webbing character has been diagnosed in counting of free phalanges (without webbing), then we have scored each free phalanges following the method described by [15] using the formula from [16] to facilitate comparisons with other species of *Mantidactylus* and most subsequent authors who published accounts on Madagascan anurans (0 = webbing reaching end of terminal phalange, 0.25 = 0.25 of terminal phalange free of webbing, 1 = terminal phalange free of webbing etc.). Extending the total sum of free phalanges for each limb is called webbing score (WS).

All genetic samples are listed in Supplemental Data with their Genbank accession numbers (see Appendix D; Appendix A), including our newly sequenced specimens and additional *Ochthomantis* samples from Genbank (see [11]). Trees were rooted using *Mantidactylus peraccae*, *M. cowanii*, *M. lugubris*, and *M.* guttulatus; species related to *Ochthomantis* [2,11]. DNA from either frozen or ethanol-preserved (70%) tissue samples was isolated using the QIAGEN DNeasy spin columns. DNA from formalin-fixed museum specimens was extracted using a modified method from [17] and precautionary steps were taken to prevent contamination [18]. To allow for the inclusion of other *Ochthomantis* sequence data from previous studies (see [2,11]), our sequencing efforts focused on the mitochondrial 16S rRNA gene. PCR amplification was performed under locus-specific parameters. All sequences were aligned using MUSCLE. BLAST searches (NCBI) were performed for each contiguous sequence to identify any potential contamination. The data set was partitioned by stems and loops (for the 16S rRNA locus) for Bayesian analysis conducted in BEAST v.2.7.5. Markov chain was run for 3 × 10^8^ generations and trees were sampled every 3000 generations. To determine that stationarity had been reached, we ensured that effective sample size of all parameters was greater than 200, using TRACER v1.6 [19]. The first 20% of trees were discarded as burn-in. The trees retained were combined to produce a maximum clade credibility tree.

The 16S rRNA clades identified from the molecular analyses were used as guide to establish congruent morphological groups that could be diagnosed using fixed morphological character states. The resulting recognized groups were thus supported by both molecular and morphological diagnostic criteria and we consider them here as species, based on these congruent dual species recognition criteria (see [20]). All recognized species also were more than 3% divergent to all other species based on the 16S rRNA gene (uncorrected distance), thus also meeting another species recognition criterion that has been widely applied in Malagasy amphibian taxonomy, e.g., [21]. All currently recognized *Ochthomantis* species were assigned to their appropriate congruent molecular and morphological group based on the character states given in their species descriptions, and directly observed by us through examining type specimens. For the unnamed congruent molecular and morphological groups, we compared their morphological diagnoses with character states in all other currently considered junior synonym *Ochthomantis* species using the same methods as described above for the senior synonyms. In cases of morphological correspondence, this was used as justification for resurrecting junior synonyms. For the remaining unnamed groups, we considered these as undescribed species. These species are formally described here. We also provide re-descriptions of species that we resurrect from junior synonymy. At the end, we also provide a key to identify the 11 species of the *Ochthomantis* subgenus.

## 3. Results

### 3.1. Molecular Analysis, and Diversity

Figure 1 shows the recovered 16S rRNA tree topology. Based on these clades and their corresponding morphological congruence, we consider the subgenus *Ochthomantis* to include 11 species. While *Mantidactylus majori* has been considered to be within *Ochthomantis* and is recovered as such in this tree, we posit that this species belongs to another subgenus according to morphological results from this study and [2]. We discuss this in more detail within the taxonomy chapter (Discussion). All eleven species of the subgenus *Ochthomantis* have >3% uncorrected *p*-distance for 16S, i.e., more than 3% divergent to each other based on the 16S rRNA gene (uncorrected distance), and are readily diagnosable based on their morphology (see Figure 1). Five species are currently recognized: *M. ambreensis*, *M. ambony*, *M. femoralis*, *M. mocquardi*, and *M. zolitschkia*; two are currently junior synonyms that require recognition as good species: *M. catalai* and *M. poissoni*; and four species are undescribed. The specimens labeled with asterisks (see Specimens Examined and Appendix A) have been sequenced for 16S and were used to estimate the gene tree through their DNA sequences and alignment (see Appendix D).

### 3.2. Redescription of Currently Recognized Species

The following five species are redescribed here, based on morphological data for newly available specimens and previously reported voucher specimens, which considerably increase the samples sizes: *M. femoralis* (124 specimens), *M. ambreensis* (46 specimens), *M. ambony* (14 specimens), *M. mocquardi* (148 specimens), and *M. zolitschka* (10 specimens) (Figure 2). We provide these redescriptions to facilitate comparison with the descriptions provided for the new and resurrected species.

#### 3.2.1. *Mantidactylus femoralis* [8]

*Rana femoralis* [8] (p. 462) (Syntypes: BMNH 1947.2.22.65–68, according to [22] (p. 26); BMNH 1947.2.22.65 designated lectotype by [10] (p. 85); *Rana flavicrus* [23] (p. 245); *Mantidactylus flavicrus* [9] (p. 450); *Mantidactylus femoralis:* [24] (p. 393); *Mantidactylus* (*Mantidactylus*) *flavicrus* [25] (p. 25); *Mantidactylus femoralis*: [26] (p. 235); *Mantidactylus (Hylobatrachus) femoralis*: [27] (p. 312); *Mantidactylus (Ochthomantis) femoralis*: [7] (p. 400); [3] (p. 3).

Lectotype: BMNH 1947.22.65, type locality: “East Betsileo”, Madagascar;

Paratype: BMNH 1947.22.66–68. same location as the lectotype;

Paralectotypes: BMNH 1947.22.66–68. same location as the lectotype.

Specimens Examined: BMNH 1947.22.65–68: East Betsileo. AMNH A23781 and A50366: Moramanga, District Moramanga, Alaotra Mangoro Region, Madagascar. AMNH A157112 adult male: Ampanasana Ankolony (14°26,2′ S 49°46,5′ E), Marojejy National Park, Andapa District, Sava Region, Madagascar, October 1998, A. Raselimanana. AMNH A167521* (RAX 6345) subadult: Analapakila Trois Lacs, District Bealanana (14°26.233′ S 48°36.696′ E, 1400 m), 12 March 2003, N. Rabibisoa and S. Mahaviasy. AMNH A167580* (RAX 2703) juvenile: Antsahatelo (13°51.588′ S 48°51.979′ E, 800 m), 6 April 2001, S. Mahaviasy, N. Rabibisoa, C. J. Raxworthy, A Razafimanantsoa, and A. Razafimanantsoa. AMNH A167581* (RAX 2761) adult male: Ramena River, Tsaratanana Reserve, Ambanja District (13°55.071′ S 48°53.179′ E; 750 m), 8 April 2001, same collectors as previous. AMNH A174623* (RAX 7196) female adult: Ankafina Tsarafidy, Ambohimasoa District, Haute Matsiatra Region, Madagascar (21°12.598′ S 47°12.874′ E, 1420 m), 16 February 2004, N. Rabibisoa, M. Randriambahiniarime, F. Ranjanaharisoa, and C. J. Raxworthy. AMNH A174627* (RAX 7523) adult male: Betampona Reserve, Toamasina District, Atsinanana Region, Madagascar (17°55.866′ S 49°12.190′ E, 350 m); 2 February 2004, N. Rabibisoa, M. Randriambahiniarime, F. Ranjanaharisoa, and C. J. Raxworthy. AMNH A174646* (RAX 8133) adult female: Manasamena, Lakato, Moramanga District, Alaotra–Mangoro Region, Madagascar (19°02.637′ S 48°20.910’, 950 m), 27 March 2004, N. Rabibisoa and N. Rakotondrazafy. AMNH A174651* (RAX 8896) juvenile: Kianjavato–Vatovavy, Ranomafana District, Vatovavy–Fitovinany Region, Madagascar (21°22.791′ S 47°52.052′ E, 150 m), 18 February 2006, N. Rabibisoa and C.J. Raxworthy. AMNH A174654* (RAX 9498) juvenile: Ambohibehivavy–Vasiana, Betafo District, Vakinankaratra Region, Madagascar (19°41.387′ S 46°06.953′ E, 850 m), 28 March 2006, N. Rabibisoa, N. Rakotondrazafy, and J. Rafanomezantsoa. AMNH A181735* (RAX 10606): Beampingaratsy Pass Anosy Mts, District Tolagnaro, Anosy Region, Madagascar (24°28.244′ S 46°53.521′ E, 520 m), Feb. 13, 2009, C.J. Raxworthy. AMNH A187128* (RAX 10901): Ambatomenaloha/Itremo, Ambatofinandrahana District, Amorin’I Mania Region, Madagascar (20°37.130′ S 46°33.347′ E, 1650 m), 19 December 2009, C.J. Raxworthy. UADBA 4517–18, 4520 (RAN 52471, 52470, 52109): Eminiminy, Andohahela National Park, Tolagnaro District, Anosy Region, Madagascar (24°37.55′ S 46°45.92′ E, 500 m), 21 October 1995, A. Raselimanana and J. B. Ramanamanjato. UADBA 20478–20479: Andriankely, Anjozorobe–Angavo National Parc, Anjozorobe District, Analamanga Region, Madagascar (18°25.225′ S 47°56,245′ E, 1250 m), 2 February 2003, M. Anjeriniana. UADBA 26118, 26268–26270 (NR 1866, 1819, 1858, 1859): Ampanatovana Lakato, Moramanga District, Alaotra–Mangoro Region, Madagascar (19°02.637′ S 48°20.912′ E, 1025 m), 29 November 2003 and 6 December 2003, N. Rabibisoa, N.A. Rakotondrazafy. UADBA 26249, 26262, 26389 (RAX7198, 7197, 7212): Ankafina Tsarafidy, Ambohimasoa District, Amoron’I Mania Region, Madagascar (21°12.598′ S 47°12.874′ E, 1150 m), 16 February 2004, N. Rabibisoa, M. Randriambahiniarime, F. Ranjanaharisoa, and C.J. Raxworthy. UADBA 26250–26251, 26263–26265, 26402 (RAX 7961, 7553, 7945, 7555, 8002, 7618): Betampona Strict Natural Reserve, Toamasina II District, Atsinanana Region, Madagascar (17°55.866′ S 49°12.190′ E, 250–450 m), 28 February 2004–17 March 2004, N. Rabibisoa, M. Randriambahiniarime, F. Ranjanaharisoa, and C. J. Raxworthy. UADBA 26252 (RAN 44674) adult male: Sahavatoy River, Andringitra National Park, Ambalavao District, Ihorombe Region, Madagascar (22°13.667′ S 47°0.217′ E, 810 m), 24 November 1993, N. Rabibisoa, A. Razafimanantsoa, and C. J. Raxworthy. UADBA 26253–26254, 26374, 26378 (RAN 45537, 45684, 45538, 44833): Rangovalo Ridge, Zahamena National Park, Ambatondrazaka District, Alaotra–Mangoro Region, Madagascar (17°40,5′ S 48°45.5′ E, 1150 m), 4–8 March 1994, J.B. Ramanamanjato, A. Raselimanana, C.J. Raxworthy, A. Razafimanantsoa, and A. Razafimanantsoa. UADBA 26255, 26380 (RAN47059, RAN47047): Andranomangoboka Ambohijanahary, Morafenobe District, Melaky Region, Madagascar (18°14.787′ S 45°21.419′ E, 730–950 m), 16 January 1995, J. B. Ramanamanjato, A. Raselimanana, C. J. Raxworthy, A. Razafimanantsoa, and A. Razafimanantsoa. UADBA 26257 (RAN 47663): Ambohimanana Tolongoina, Ikongo District, Vatovavy–Fitovavy Region, Madagascar (21°28.557′ S 47°33.759′ E, 600 m), 9 February 1995, J. B. Ramanamanjato, A. Raselimanana, C. J. Raxworthy, A. Razafimanantsoa, and A Razafimanantsoa. UADBA 26258 (RAN 47953): Sahamalio, Isalo National Park, Ranohira District, Ihorombe Region, Madagascar (22°26.315′ S 45°15.648′ E, 700 m), 18 February 1995, C. J. Raxworthy, A. Raselimanana, J. B. Ramanamanjato, A Razafimanantsoa, and A Razafimanantsoa. UADBA 26259, 26284, 26260–26261 (RAN 47995, 47996, 48219, 48221): Canyon Singe, Isalo National Park, Ranohira District, Ihorombe Region, Madagascar (22°29.138′ S 45°23.086′ E, 600 m), 20 February 1995, same collectors as previous. UADBA 26266, 26385 (RAX 8157, 8158): Manasamena Lakato, Moramanga District, Alaotra Mangoro Region, Madagascar (19°02.637′ S 48°20.910′ E, 950 m), 27 March 2004, N. Rabibisoa and N. Rakotondrazafy. UADBA 26267 (RAN 45891): Namarafana, Zahamena National Park, Ambatondrazaka District, Alaotra Mangoro Region, Madagascar (17°44′ S 48°58.5′ E, 420 m), 16 March 1994, J. B. Ramanamanjato, A. Raselimanana, C. J. Raxworthy, A. Razafimanantsoa, and A. Razafimanantsoa. UADBA 26281, 26376 (RAN 47280, 47248): Doany Ambohijanahary, Morafenobe District, Melaky Region, Madagascar (18°17.297′ S 45°33.289′ E, 1220 m), 24 January 1995, same collectors as previous. UADBA 26282 (RAN 47580) adult female: Ambohitantely Special Reserve, Ankazobe District, Analamanga Region, Madagascar (18°11.158′ S 47°16.757′ E, 1580 m), 2 February 1995, same collectors as previous. UADBA 26363 (JB 127) adult female: Andranomay, Anjozorobe District, Analamanga Region, Madagascar (18°28.8′ S 47°57′ E, 1300 m), 12 December 1996, A. Raselimanana. UADBA 26377 (RAN 47631) same condition as UADBA 26257 except date of collection Feb. 8, 1995. UADBA 26375, 26379 (RAN 45395, 45364): Volontsagana River, Zahamena National Park, Ambatondrazaka District, Alaotra–Mangoro Ragion (17°42′ S 48°46′ E, 850 m), 28 February 1994, J. B. Ramanamanjato, A. Raselimanana, C. J. Raxworthy, A. Razafimanantsoa, and A. Razafimanantsoa. UADBA RAX10955*: Mandraka, Manjakandriana District, Analamanga Region, Madagascar (18°57.733′ S 47°55.050′ E, 1140 m), 18 March 2006, N. Rabibisoa, and J. Rafanomezantsoa. UMMZ 212835 (RAN 39921) adult male: Manantenina River, Marojezy National Park, Andapa District, Sava Region, Madagascar (14°26′ S 49°46′ E, 700 m), 25 November 1992, R. A. Nussbaum, A. Razafimanantsoa, A Razafimanantsoa, and C J Raxworthy. UMMZ 197651* (RAN 35289): Manantantely Forest, District Tolagnaro, Anosy Region (24°39′ S 46°55.083′ E, 125 m), 6 November 1990, J.B. Ramananjato, A Raselimanana, RA Naussbaum, and C J Raxworthy.

Additional specimens examined: see Appendix C.

Diagnosis: A medium- to large-sized *Ochthomantis* (adult male SVL 31–43.4 mm; adult female SVL 43.0–62.4 mm); tibiotarsal articulation beyond nostrils (rarely between eye and nostril); 1.5–2 free phalanges on the internal edge of toe 4; width of digit terminal disc ≥ 1.70 disc base; white stripe along the superior lip and prominent yellow patch in the inguinal region. Distinguished from other subgenus species by the following combination of characters: from *M. ambreensis* and *M. ambony* by yellow line or patch in the inguinal region; *M. poissoni* by absence under the eye of large white spot or partly fused white spots on upper lip; *M. mocquardi, M. catalai*, *M*. *olgae* n. sp., and *M. tavaratra* n. sp. by the presence of 1.5–2 free phalanges on the internal edge of toe 4, yellow patch in the inguinal region, and tibiotarsal articulation beyond eye; *M. zolitschka* by its large size (SVL ≥ 33 mm) and yellow patch in inguinal region; *M*. *danieli* n. sp. by the yellow patch in inguinal region and width of the digit terminal disc ≥ 1.70 base of disc; *M. macrotympanum* n. sp. by smaller adult male SVL (<60 mm). Character diagnostics are summarized in Table 1 and Table 2.

Description of reference specimen UADBA 19662 (NR 1724): adult male (SVL = 37.45 mm) in excellent state of preservation. Measurements are presented in Appendix A. Snout tip pointed in dorsal and lateral views and triangular in ventral view. Snout tip with 1.75 mm ventral extension beyond the mouth. Head 1.44 times longer than width. Head length 0.48 times SVL. Canthus rostralis weakly evident. Tympanum diameter 0.88 times eye. Ovoid tympanum distinct to supratympanic fold and which runs towards above forearm and shoulder girdle articulation. Internarial distance 0.28 times head width. Tongue ovoid anteriorly and bifid posteriorly. Round nostrils with lateral aperture. Eye to nostril distance 1.45 times nostril to snout distance. Forearm length 0.50 times SVL. Hand length (including discs) 0.30 times SVL. Fingers without webbing. Inner and outer tubercle metacarpals are very developed in granule-like patterns. Fingers without webbing. Relative finger length 1 < 2 < 4 < 3. Digits with large terminal discs (widest part twice of basal disc width). Tibiotarsal articulation beyond snout tip. Lateral metatarsal separated. Hindlimb 1.90 times SVL. Thigh length 0.95 times tibia length. Foot including tarsus 0.82 times SVL. Inner metatarsal tubercle shield-like pattern at the base of toe 1 (length 1.5 mm). Outer metatarsal tubercle with small granule. Toes with extending webbing and webbing formula: 1 (1), 2i (1), 2e (0.5), 3i (1), 3e (0.5), 4i (2), 4e (1.5), 5 (0). Total sum of free phalanges 7.50: Relative toe length 1 < 2 < 3 < 5 < 4. The lateral surface of the body with small granules and the dorsal surface of the body is almost without granules. Oblong femoral glands slightly developed and its medio-proximal area with pores surrounded by many granules, giving them a crater-like pattern. Internally, femoral glands type 3 [2,28]. In the preservative, the dorsal surface of the head, body, and limbs with dark brown color and a thick pale brown vertebral line running from the snout tip to the anal pore. Upper lip with white band on its lower edge. Lower lip with white spot. The ventral surface of the body with homogeneous pale brown color and darker brown spot pigments, but spots become lighter or less dense on the belly. Throat with dark brown parallel stripes. The inguinal region with prominent pale yellow oblique line. Outer sides of hindlimb and forelimb with darker brown transversal bands.

Variation: Morphometric variation is summarized in Appendix A. Sexual dimorphism is evident: males have smaller SVL than females (33–43 mm vs. 43–62 mm), shorter tibia, larger toe 3, and the head relatively smaller. Ratio td/ed larger in males than in females (0.69–1.00 vs. 0.52–0.76). The femoral glands are more swollen in males and smaller and more circular in females. The cloacal area in females has black spots. The free phalanges on toes vary: toe 1 (0.25–1), internal edge of the toe 2 (1–1.50), external edge of the toe 2 (0–1) internal edge of the toe 3 (1–1.75), external edge of the toe 3 (0–1), internal edge of the toe 4 (1.50–2), external edge of the toe 4 (1–2), and in toe 5 (0–0.75).

Coloration in life: The iris has a golden ring on its outer area. A dark brown transversal bar is present between the eyes. A dorsal surface of the body may have or not have a pale yellow vertebral line. The inguinal region has a distinct oblique and large yellow patch, sometimes, it extends onto the ventral side. The ventral surface of the body is often pale brown on the anterior parts and brown yellowish color on the belly. The throat sometimes has a pair of dark brown parallel stripes. The inner surface of the thigh has a brown color and its outer part has yellow spots. Hindlimbs have alternate transversal bands, brown and black.

Habits: semi-aquatic rainforest species living close to rivers or small streams with slow-flowing water and rarely, either observed outside the rainforest or along the streams close to relict forests (e.g., Tsarafidy). This species was found on rocks, leaves, and branches or on the ground along river banks between 9.00 am and 23.30 pm. However, it is more active at night. During the day, it can be found hidden in holes and rock crevices along rivers. At night, the females are generally found sitting on shrubs living along streams and more rarely observed on the rocks far from the riverbank. Males can be found in all areas along rivers, except on branches and plant stems. The vertical distribution of individuals on shrubs might be different between the sexes: we observed males up to 1 m above the ground and females up to 2 m. Individuals prefer streams with stream beds between 1–10 m in width and water depth between 10–150 cm. Eggs are laid in masses outside the water, either on a leaf or a branch overhanging of the stream or the river. The tadpoles live in the calm water far from torrents. The period of reproduction is during the cold winter season (e.g., June and July in Mantadia). The calls of males are low and hardly audible. In the Moramanga Region, metamorphosis appears to occur from September to October. At Andranomanamponga in August 2002, the tadpoles of different development stages were observed in calm and transparent water, and these tadpoles’ habitats were protected by rocks. These pools have surface water between 1–3 m^2^ and a water depth between 20–100 cm. When disturbed, the tadpoles swim quickly and obliquely to hide in mud under dead leaves or retreat into rock crevices.

Distribution: Eastern and northern rainforests, including the highland of Madagascar with an elevation range between 230 to 1600 m above sea level. The species occurs as far north as the Sambirano Region (14° S) and as far south as the Anosy Mountains (25° S) (Figure 3, Appendix A).

Comments: Our morphological description agrees with the *M. femoralis* shown by [29] (Figure 2, p. 247) from Antoetra, and our molecular analyses groups all our *M. femoralis* samples (Appendix A) with the *M. femoralis* samples reported by [10]: AY324815 (FGMV 2001.155) AY324817 (FGMV 2002.56); and [11]: HQ610845 (ZSN 1630/2007), HQ610847 (ZSM 1643/2007), HQ610913–610914 (FGZC 271, ZCMV 370) and HQ 610916–61091618 (ZCMV 464, 937, 5874). *M. flavicrus* (with type locality “Madagascar”) remains a junior synonym of *M. femoralis*.

#### 3.2.2. *Mantidactylus ambreensis* [30]

*Mantidactylus ambreensis* [30] (p. 127). Validated as bona species by [31] (p. 67); *Mantidactylus (Ochthomantis) ambreensis:* [7] (p. 400); [3] (p. 3).

*M. ambreensis* was synonymized with *M. femoralis* by [22], and later recognized again as a good species by [10,31].

Holotype: MNHN 1893.241: Montagne d’Ambre, Madagascar.

Specimens examined: MNHN 1893.241 adult female: Montagne d’Ambre, Madagascar. AMNH A50521: Analalava District, Sofia Region, Madagascar, 21 January 1971. AMNH A167482* (RAX 2373) adult male: Mandrizavona, Tsaratanana Reserve, Ambanja District, Diana Region, Madagascar (13°48.043′ S 48°44.78′ E, 650 m), 12 January 2001, C. J. Raxworthy, A. Razafimanantsoa and A. Razafimanantsoa. AMNH A167485 (RAX 2468) adult male: Besahona, Tsaratanana Reserve, Ambanja District, Diana Region, Madagascar (13°54.372′ S 48°52.425′ E, 550–750 m), 25 January 2001, same collectors as AMNH A167482*. AMNH A167486 (RAX 2720): Antsahatelo, Tsaratanana Reserve, Ambanja District (13°51.588′ S 48°51,979′ E, 700 m), 7 April 2001, S. Mahaviasy, N. Rabibisoa, C. J. Raxworthy, A. Razafimanantsoa, and A. Razafimanantsoa. AMNH A167499–167500 (RAX 3289, 3330, 3343): Antsaravy, RNI Tsaratanana, Ambanja District, Diana Region, Madagascar (13°55.560′ S 48°54.353′ E, 1150 m), 20 April 2001, same collectors as AMNH A167486. AMNH A167501*–167502*–04 (RAX 6552, 6557, 6584, 6558): Irony Relict Forest Camp, Bealanana District, Sofia Region, Madagascar (14°44.937′ S 48°29.449′ E, 930 m).

Collected 1 April 2003, S. Mahaviasy, N. Rabibisoa, N. Rakotondrazafy, and C. J. Raxworthy.AMNH A 167565* (RAX 3203) juvenile: Antsaravy, RNI Tsaratanana, District Ambanja, Diana Region, Madagascar (13°55.560′ S 48°54.353′ E, 1150 m), 16 April 2001, N Rabibisoa, S. Mahaviasy, A. Razafimanantsoa, and A. Razafimanantsoa. AMNH A174618* (RAX 9589) adult male: Ambohibola forest, Tsaratanana District, Betsiboka Region, Madagascar (16°38.358′ S 47°26.165′ E, 300 m), 10 April 2006, N. Rabibisoa, and C. J. Raxworthy. UADBA 3714 (RAN 38503) adult male: Fitsahana Atomboka River, Montagne d’Ambre, Antsiranana District, Diana Region, Madagascar, (12°29.2′ S 49°10,3′ S, 1150 m), 25 December 1991, by C. J. Raxworthy, A. Raselimanana, J. B. Ramanamanjato. UADBA 5647–5650 (RAN 54050, 54105, 54130, 54054): Irony Relict Forest Camp, Bealanana District, Sofia Region, Madagascar (14°44.937′ S 48°29.449′ E, 930 m), 21–24 February 1996, C. J. Raxworthy, A. Razafimanantsoa, A. Razafimanantsoa. UADBA 7222–7224: Benavony, Ambanja District, Diana Region, Madagascar, 200 m, Mar. 21, 1994, F. Glaw, N. Rabibisoa, and O. Ramilison. UADBA 8393, 8396, 8414 (RAX 2796, 2875, 3000): Ramena River Camp, Ambanja District, Diana Region, Madagascar (13°55.071′ S 48°53.179′ E, 730–750 m), 9–13 April 2001, Mahaviasy, N. Rabibisoa, C. J. Raxworthy, A. Razafimanantsoa, and A. Razafimanantsoa. UADBA 8394, 8411, 8416 (RAX 2530, 2528, 2514): Besahona, Ambanja District, Diana Ragion (13°54.372′ S 48°52.425′ E, 550 m), 27 January 2001, A. Razafimanantsoa, and Razafimanantsoa. UADBA 8395, 8398, 8412, 8415 (RAX 2661–2658): Nirhy’s cascade Camp Analabe, Tsaratanana Reserve, Ambanja District, Diana Region, Madagascar (13°51.023′ S 48°47.902′ E, 760 m), 5 April 2001, same collectors as UADBA 8393. UADBA 8397, 8413 (RAX 2557, 2401): Mandrizavona, Ambanja District, Diana Region, Madagascar (13°48.043′ S 48°44.78′ E, 450 m), 14 January and 30 January 2001, A. Razafimanantsoa, and A. Razafimanatsoa. UADBA 8399 (RAX 2554) adult female: Betaindambo, Ambanja District, Diana Region, Madagascar (13°51.932′ S 48°49.189′ E 550 m), 30 January 2001, A. Razafimanantsoa, and A. Razafimanatsoa. UADBA 8401 (RAX 2702) adult female: Antsahatelo, Tsaratanana Reserve, Ambanja District, Diana Region, Madagascar (13°51.588′ S 48°51.979′ E, 800 m), 6 April 2001, same collectors as UADBA 8393. UADBA 8408, 8410 (RAX 3195, 3236): Antsaravy Valley Camp, Tsaratanana Reserve, Ambanja District, Diana Region, Madagascar (13°55.560′ S 48°54.353′ E, 1150 m), 15 April and 17 April 2001, S. Mahaviasy, N. Rabibisoa, A. Razafimantsoa, and A. Razafimanantsoa. UADBA 9056–9057 (NR 548, 547): Les Rousettes Camp, Montagne d’Ambre National Park, Antsiranana District, Diana Region, Madagascar (12°31′ S 49°10′ E, 1000 m), 3 March 1996, N. Rabibisoa, D. Rakotomalala, and O. Ramilison. UADBA 26120, 26222–26224, 26271–26272 (RAX 6556, 6586, 6553, 6555, 6554, 6559): Irony Relict Forest Camp, District Antsohihy, Sofia Region, Madagascar (14°44.937′ S 48°29.449′ E, 930) and (14°45.140′ S 48°29.690′ E, 950 m), 1–2 April 2003, S. Mahaviasy, N. Rabibisoa, N. Rakotondrazafy, and C. J. Raxworthy. UMMZ 212435* (RAN 38501): Antomboka River, Montagne d’Ambre, Antsiranana District, Diana Region, Madagascar (12°32.3′ S 49°10′ S, 650 m), 25 December 1991, J.B. Ramanamanjato, A. Raselimanana, and C.J. Raxworthy.

Diagnosis: A relatively small-sized *Ochthomantis* (adult male SVL 33.3–39 mm; adult female 38.2–42 mm) with a distinct lateral white strip running along lateral surfaces of head and body, from snout tip to inguinal area. Distinguished from *M. ambony* by its larger size and from the remaining species by the presence of a sharply white lateral stripe well-defined. Character diagnostics are summarized in Table 1 and Table 2.

Description of the reference specimen UADBA 8393 (RAX 2796): Adult male (SVL = 39.00 mm) in excellent state of preservation. Measurements are presented in Appendix A. In dorsal view and lateral view, the snout tip is pointed. Snout tip with 2.0 mm ventral extension beyond the mouth. In the dorsal view, the head is clearly longer than large, i.e., 1.38 times longer than wide. Head length 0.44 times SVL. Canthus rostralis indistinct. Loreal area concave. Tympanum diameter 1.04 times eye. Round tympanum in contact with a supratympanic fold along their borders except in its posterior part. Supratympanic fold running posteriorly to one point of three large granules above the upper arm and shoulder girdle articulation. Dark tympanum with a small notch in its median superior area. Internarial distance 0.27 times head width. The tongue is ovoid anteriorly and bifid in its posterior part. Round nostrils with lateral aperture. Eye to nostril distance 1.97 times nostril to snout distance. Forearm length 0.48 times SVL. Hand length (including discs) 0.17 times SVL. The inner metacarpal is not obvious and the outer metacarpal has a flattened granule. Fingers without webbing. Relative finger length 1 < 2 < 4 < 3. Digits with large terminal discs (the widest part is twice the width of the basal disc). Tibiotarsal articulation between eye and nostril. Lateral metatarsal separated. Hindlimb 1.77 times SVL. Thigh length 0.97 times tibia length. Foot including tarsus 0.74 times SVL. Inner metatarsal tubercle not obvious (length 1.80 mm) at base of toe 1. Outer metatarsal tubercle absent. Toe with relative extending webbing and webbing formula: 1 (1) 2i (1) 2e (0.50), 3i (1.25) 3e (0.50) 4i (1.50) 4e (1.50) 5 (0.50). The total sum of free phalanges 7.75. The relative toe length is 1 < 2 < 3 < 5 < 4. Lateral surfaces of the body, belly, and sacral areas with very small granules. Oblong femoral glands are well-developed and its medio–proximal area with a pore surrounded by many granules, giving it a crater-like pattern. Internally, femoral gland type 3 [2,28]. In the preservative, the dorsal surface of the head, body, and limbs is dark in color. The lateral surface of the body is less dark than the dorsal surface of the body. Lips with white band colors continue along the lateral surface of the body to above hindlimb articulation. Dark iris surrounded by a white ring. The ventral surface of the body is whitish with dark marbling except in the belly region. Throat with dark-brown parallel bands. Ventral surface of the forelimb with some dark spots. Thigh with large dark spots except on the femoral gland region. Hind and forelimbs with alternate and transverse dorsal bands, dark and light gray colors.

Variation: Morphometric variation is summarized in Appendix A. Sexual dimorphism is evident: males have smaller SVL (33.3–39 mm versus 38.2–42 mm), relatively longer tibia and feet, and shorter toe 3; ratio td/ed larger in males than females (0.80–1.16 vs. 0.52–0.79); femoral glands more swollen in males and smaller and more circular in females. The free phalanges on toes vary: toe 1 (0–1), internal edge of toe 2 (1), external edge of toe 2 (0–0.50), internal edge of toe 3 (1–1.50), external edge of toe 3 (0–1), internal edge of toe 4 (1–2), external edge of toe 4 (1–2), and in toe 5 (0–1). Small skin granules may be present or not on the dorsal surface of the body and above the eyes. Some individuals with granules on the sacral area.

Coloration in life: The iris has a gold ring on its outer area. A dark crossbar may be present between the eyes. There is no vertebral line on the dorsum. A white or yellow band is running along the flank. The dorsal surfaces of the head, body, and limbs may be dark green, brown, or grayish brown in color. The ventral surfaces of the body are usually mottled brown, but for some individuals, there is almost no mottling. Some individuals have a diverged pair of short longitudinal dark bands on the throat and this fades on the thorax. Large and round dark spots may also be present on the throat and thorax.

Habits: semi-aquatic forest species living next to flowing streams or rivers generally and close to rocks. This species was found between 9.00–23.00 h, but it is rather nocturnal than diurnal frogs. At night, females rested on leaves rather than on branches, and during the day, we observed them sometimes resting on the banks of the river and very rarely on the ground. For males, during the day they rested on banks and at night on leaves, and sometimes we can find them on rocks. The vertical distribution of individuals on shrubs is different between both sexes: males between 10–200 cm, whereas females between 100–200 cm. Distinguished from *M. femoralis*, this taxon prefers rivers to small streams. At Montagne d’Ambre (Station les Roussettes), it is observed near the irrigation canal. Calling males were heard in March in the afternoon from the ground along the forest brook [7]. A clutch consisting of about 100–120 eggs was found deposited on a rock edge close to a calm and shallow stream in April 2006 at Vohibola Tsaratanana and those eggs were hatched after a couple of days in a plastic bag. The egg diameter is about 2–3 mm.

Distribution: species of low and mid-altitude forests of Northwestern and Northern Madagascar, with elevational ranges between 200 to 1150 m. This species is found as far north as Montagne d’Ambre and across humid forests in the northern highlands such as Manongarivo, Tsaratanana, Andramanalana, and Sorata; and extending to the northern limit of the High Plateau at Irony River and Ambohibola Forest (Figure 4, Appendix A).

Comments: Our molecular analyses group all our *M. ambreensis* samples with the *M. ambreensis* samples reported by [10]: AY324822 (ZSM 492/2000); [11]: HQ610870 (FGMV 2002.1950); and by the holotype MNHN 1893.241 (sequence accession MT982173) [5].

#### 3.2.3. *Mantidactylus ambony* [5]

Holotype: ZSM 2078/2007 (FGZC 1039): adult female, Montagne dʼAmbre National Park, Antsiranana District (12.5280° S, 049.1720° E, 1050 m), 24 February 2007, F. Glaw, P. Bora, H. Enting, J. Köhler, and A. Knoll [5].

Specimens Examined: AMNH A167487 (RAX 2758): Antsahatelo, Tsaratanana Reserve, Ambanja District (13°51.588′ S 48°51,979′ E, 700 m), 7 April 2001, S. Mahaviasy, N. Rabibisoa, C. J. Raxworthy, A. Razafimanantsoa, and A. Razafimanantsoa. AMNH A167490–16795, 167497 (RAX 2816, 2872–2874, 2876, 2932, 2970): Ramena, Ambanja District, Diana Region, Madagascar (13°55,071′ S 48°53,179′ E, 730–750 m), 11 April 2001 except AMNH A167497 12 April 2001, S. Mahaviasy, N. Rabibisoa, C. J. Raxworthy, A. Razafimanantsoa and A. Razafimanantsoa. AMNH A167498 (RAX 3289): Antsaravy, RNI Tsaratanana, Ambanja District, Diana Region, Madagascar (13°55.560′ S 48°54.353′ E, 1150 m), 20 April 2001 18 April 2001, S. Mahaviasy, N. Rabibisoa, C. J. Raxworthy, A. Razafimanantsoa, and A. Razafimanantsoa. UADBA 7225–7226: Les Roussetes Camp, Montagne d’Ambre National Park, Antsiranana District, Diana Region, Madagascar (12°31′ S 49°10′ E, 1000 m), Feb. 02, 1994, F. Glaw, N. Rabibisoa, and O. Ramilison. UADBA 8406 (RAX 2871): Ramena River Camp, Ambanja District, Diana Region, Madagascar (13°55.071′ S 48°53.179′ E, 730–750 m), 9–13 April 2001, Mahaviasy, N. Rabibisoa, C. J. Raxworthy, A. Razafimanantsoa, and A. Razafimanantsoa. UADBA 9058 (NR 564): Les Rousettes Camp, Montagne d’Ambre National Park, Antsiranana District, Diana Region, Madagascar (12°31′ S 49°10′ E, 1000 m), 3 March 1996, N. Rabibisoa, D. Rakotomalala, and O. Ramilison. UADBA 5726 (RAN 51382): 34 km South from Maevatanana, Maevatanana District, Betsiboka Region, Madagascar (17°09.092′ S 46°51.365′ E, 350 m), 24 January 1996, A. Raselimanana, C. J. Raxworthy, A. Razafimanantsoa, and A. Razafimanantsoa. UMMZ 212426* (RAN 38009): Antomboka River, Montagne d’Ambre, Antsiranana District, Diana Region, Madagascar (12°32.3′ S 49°10′ S, 1150 m), 15 November 1991, J.B. Ramanamanjato, A. Raselimanana, and C.J. Raxworthy.

Diagnosis: A small sized *Ochthomantis* (adult male SVL 30.0–31.8 mm; adult female 34.0–37.9 mm) with sharply white to yellow stripe, well-defined along the lateral head and lateral surface of body, running from snout tip to groin area. Distinguished from *M. ambreensis* by its smaller size and from the remaining species by the presence of a sharply white to yellow well-defined lateral stripe. Character diagnostics in Table 1 and Table 2.

Description of reference specimen: UADBA 8406 (RAX 2871): Adult male (SVL = 31.8 mm) in excellent state of preservation. Measurements are presented in Appendix A. In the dorsal view and lateral view, the snout tip relatively pointed. Snout tip with 1.7 mm ventral extension beyond mouth. In the dorsal view, the head is clearly longer than large (i.e., 1.35 times longer than wide). Head length 0.47 times SVL. Canthus rostralis indistinct. Loreal area concave. Tympanum diameter 1.19 times eye. The round tympanum is in contact with the supratympanic fold along their borders and running posteriorly above towards the upper arm and shoulder girdle articulation. Large and dark tympanum with a small notch in its median superior area. Internarial distance 0.30 times head width. Tongue ovoid anteriorly and bifid in posterior part. Round nostrils with lateral aperture. Eye to nostril distance is 1.40 times nostril to snout distance. Forearm length 0.47 times SVL. Hand length (including discs) 0.19 times SVL. The inner metacarpal is not obvious and the outer metacarpal has obvious granule. Fingers are without webbing. The relative finger length is 1 < 2 < 4 < 3. Digits with large terminal discs (widest part twice of basal disc width). Tibiotarsal articulation between eye and nostril. Lateral metatarsal separated. Hindlimb 1.69 times SVL. Thigh length 0.90 times the tibia length. Foot including tarsus 0.73 times the SVL. Inner metatarsal tubercle not obvious (length 1.35 mm) at base of toe 1. Outer metatarsal tubercle absent. Toe with less extensive webbing and webbing formula: 1 (1) 2i (1) 2e (0.50), 3i (1.25) 3e (1) 4i (2) 4e (1.50) 5 (0.50), and total sum of free phalanges 8.75. Relative toe length 1 < 2 < 3 < 5 < 4. The surfaces of the body: flank, belly, and sacral areas with very small granules and dorsal with obvious granules. Oblong femoral glands are well-developed and pores in its medio-proximal area are surrounded by many granules, giving it a crater-like pattern. Internally, femoral gland type 3 [2,28]. In preservative, the dorsal surface of the head, body, and limbs has a dark color. The lateral surface of the body is less dark than the dorsal surface of the body. Lips with white band colors run along the lateral surface of the body to the hindlimb insertion. Dark iris surrounded by a white ring. The ventral surface of the body is whitish with dark marbling except in the belly region. Throat with dark brown parallel bands. Forelimb with some dark spots on its ventral surface. Thigh with large dark spots except in the femoral gland region. Fore and hindlimbs with dorsal transverse bands, alternating dark and light gray colors.

Variation: Morphometric variation is summarized in Appendix A. Sexual dimorphism is evident: males have smaller SVL (30–31.8 mm vs. 34–37.9 mm), relatively longer tibia and feet, and shorter toe 3; the ratio td/ed larger in males than females (0.93–1.33 vs. 0.59–0.79); the femoral glands are more swollen in males, but smaller and more circular in females. The free phalanges on toes vary: toe 1 (0.50–1), internal edge of toe 2 (1–1.25), external edge of toe 2 (0–0.75), internal edge of toe 3 (1.25–1.50), external edge of toe 3 (0.25–1), internal edge of toe 4 (1.50–2), external edge of toe 4 (1–2), and in toe 5 (0–1). Dorsal skin above the eyes with obvious granules. Some individuals have some granules in their sacral area.

Coloration in life: The iris has a gold ring on its outer area. A dark crossbar may be present between the eyes. No vertebral line on dorsum. A white or yellow band runs along the flank. The dorsal surfaces of the head, body, and limbs may be dark green or brown in color. The ventral surfaces are usually mottled brown. All individuals have a diverged pair of short longitudinal dark bands on the throat and fade on the thorax. Large and round dark spots were absent on the throat and thorax.

Habits: semi-aquatic forest species living next to quiet rivers with rocks. It was observed between 11.00–20.00 h. Like *M. ambreensis*, this taxon prefers rivers over small streams. At Montagne d’Ambre (Station les Roussettes), it was observed near the irrigation canals. According to [5], this is a rheophilous species and frequently terrestrial by day, sitting on the ground, on rocks, wood, lichen, or hiding under rocks. At night, often observed on perches above the water, once at 2 m height, and sitting on substrates like leaves, rocks, dead wood, and plant stems.

Distribution: Species of low and mid-altitude forests of Northwestern and Northern Madagascar, with an elevational range between 300 to 1150 m above sea level. It was observed from a relict deciduous forest, south of Maevatanana, to a humid forest at Montagne d’Ambre, extreme Northern Madagascar. All of the sites for their presence are presented in Figure 4 and Appendix A.

Comments: This species occurs only at Montagne d’Ambre [5]. Even though our single molecular state this analysis, the specimens that we have recorded outside Montagne d’Ambre based on morphological characters only should be considered too, according to character diagnosis from [5]. This is why we have grouped all our *M. ambony* samples (molecular and morphological characters) with the *M. ambony* samples reported by [5]: holotype ZSM 2078/2007 (FGZC 1039).

#### 3.2.4. *Mantidactylus mocquardi* [32]

*Mantidactylus mocquardi* [32]: 359 (Holotype MNHN 1929.207, according to [33]: 50, secondary homonym of *Rhacophurus mocquardi* [34]; *Mantidactylus (Mantidactylus) mocquardi*: [25]: 37; *Mantidactylus (Hylobatrachus) mocquardi*: [27]: 312; *Mantidactylus (Ochthomantis) mocquardi:* [7]: 400, [3]: 3.

Holotype: MNHN 1929.207 collected in Rogez, Moramanga District, Alaotra–Mangoro Region, Madagascar.

Specimens examined: AMNH A157111 (APR 234) adult female: Ampanasana Ankolony, Marojejy National Park, Andapa District, Sava Region, Madagascar (14°26.2′ S 49°46.5′ E, 1300 m), November 1998, A. Raselimanana and D. Rakotomalala. AMNH A157118–157119 (APR 351, 354) adult male and female: Andapimbazaha, Marojejy National Park, Andapa District, Sava Region, Madagascar (14°26′ S 49°46.7′ E, 850 m), the same date and collector as AMNH A157111. AMNH A167583 (RAX3806) adult female and AMNH A167585, 167587* (RAX 3903, 3915) adult males: Bezavona, Vohémar District, Sava Region, Madagascar (13°31.962′ S 49°51.954′ E, 350 m), 2 February and 8 February 2002, S. Mahaviasy, N. Rabibisoa, and C. J. Raxworthy. AMNH A167588 (RAX 4687) adult male: Ankitsika, Vohemar District, Sava Region, Madagascar (13°52′20.6″ S 49°47’02.7″ E, 650 m), 22 March 2002, N. Rabibisoa and S. Mahaviasy. AMNH A167589 (RAX 5298) adult female and AMNH A167597 (RAX 5297) adult male: Sorata, Vohémar District, Sava Region, Madagascar (13º41.986′ S 49º26.687′ E, 980 m), 22 April 2002, S. Mahaviasy, N. Rabibisoa, C. J. Raxworthy. AMNH A174621* (RAX 3669) adult female: Ambolokopatrika, Andapa District, Sava Region, Madagascar (14°32’18.1″ S 49°26’14.6″ E, 850 m), 29 November 2001, S. Mahaviasy, N. Rabibisoa, N. Rakotondrazafy, A. Razafimanantsoa and A. Razafimanantsoa. AMNH A174622* (RAX 4670): Ankitsika, Vohemar District, Sava Region, Madagascar (13°52’20.6″ S 49°47’02.7″ E, 650 m), 22 March 2002, N. Rabibisoa, S. Mahaviasy, and N. Rakotondrazafy. AMNH A174628* (RAX 7524): Betampona Strict Reserve, Toamasina II District, Atsinanana Region, Madagascar (17°54.858′ S 49°12.474′ E 350 m), Feb 28, 2004, C. J. Raxworthy, N. Rabibisoa, M. Randriambahiniharime, and F. Ranjanaharisoa. AMNH A174652* (RAX 9022): Ambodiriana, Soanierana Ivongo District, Analanjirofo Region, Madagascar (16°40.469′ S 49°42,167′E, 100 m), 3 March 2006, N. Rabibisoa. UADBA 7769 (MRJ 107) adult female: Ampanasantongotra, Marojejy National Park, Andapa District, Sava Region, Madagascar (14°26′ S 49°46.5′ E, 350 m), 10 October 1994, N. Rabibisoa, J. B. Ramanamanjato, and O. Ramilison. UADBA 8118 (NR 285) adult female: Anjanaharibe–Sud Special Reserve, Andapa District, Sava Region, Madagascar (14°44.5′ S 49°26.5′ E, 1550 m), Nov. 11, 1994, N. Rabibisoa. UADBA 12312–12313 (NR 1371–1372) adult females: Sandranantitra, Toamasina District, Atsinanana Region, Madagascar (18°2.9′ S 49°5.5′ E, 450 m), 10 January 1999, J. Randrianirina and J. Razafimanantsoa. UADBA 19596 (RAX 3680) adult male: Ambolokopatrika River, Andapa District, Sava Region, Madagascar (14°32’18.1″ S 49°26’14.6″ E, 875 m), 30 November 2001, S. Mahaviasy, N. Rabibisoa, C. J. Raxworthy, A. Razafimanantsoa and A. Razafimanantsoa. UADBA 19647* (RAX 3641) adult female: Ambolokopatrika, Andapa District, Sava Region, Madagascar (14°32’18.1″ S 49°26’14.6″ E, 850 m), 29 November 2001, the same collectors as UADBA 19596. UADBA 26238 (RAX 8021) adult female, and UADBA 26287, 26290 (RAX 7539, 8036) adult males: Betampona Strict Natural Reserve, Toamasina District, Atsinanana Region, Madagascar (17°54.858′ S 49°12.474′ E, 390–450 m), 8–18 March 2004, C. J. Raxworthy, N. Rabibisoa, M. Randriambahiniharime, and F. Ranjanaharisoa. UADBA 26240, 26242 (RAN 45476, 45363) adult females, and UADBA 26298 (RAN 45370) adult male: Rangovalo, Zahamena National Park, Fenoarivo Atsinanana District, Analanjirofo Region, Madagascar (17°40.5′ S 48°45.5′ E, and 17°42′ S 48°46′ E, 850–1150 m), 28 February 1994–3 March 1994, F. Rabemananjara, J. B. Ramanamanjato, A. Raselimanana, A. Ravoninjatovo, C. J. Raxworthy, J. Razafimanantsoa, A. Razafimanantsoa, and A. Razafimanantsoa. UADBA 26283 (RAN 47954) adult female: Sahamalio, Isalo National Park, Ihorombe Region, Madagascar, (22°26.315′ S 45°15.648′ E, 700 m), 18 February 1995, J. B. Ramanamanjato, A. Raselimanana, C. J. Raxworthy, A. Razafimanantsoa, and A. Razafimanantsoa. UMMZ 212824* (RAN 37992) juvenile: Manantenina River, close to Marojejy National Park, Andapa District, Sava Region, Madagascar (14°26′ S 49°46′ E, 600 m), 16 November 1992, R. A. Nussbaum, C. J. Raxworthy, A. Razafimanantsoa, and A. Razafimanantsoa. UMMZ 21235 (RAN 39291) adult male: Ambalafary, Ambanja District, Diana Region, Madagascar (14°04′ S 48°17′ E, 250 m), 24 february 1992, C. J. Raxworthy, A. Raselimanana, J. B. Ramanamanjato, A. Razafimanantsoa, and A. Razafimanantsoa; UMMZ 212881* (RAN 42737) juvenile: Ankavanana River, Masoala National Park, Antalaha District, Sava Region, Madagascar (15°18,5′ S/50°14′ E, 70–100 m), 12 January 1993, C. J. Raxworthy, A. Razafimanantsoa, and A. Razafimanantsoa.

For other specimens examined: see Appendix C.

Diagnosis: A medium to large-sized *Ochthomantis* species (adult male 36–48 mm; adult female 44–65 mm); tibiotarsal articulation between eye and nostril but sometimes beyond nostril; body dark brown or black color (black in preservative) with few white spots scattered on the lateral body; the upper lip of mouth paler brown with dark brown spots and densely spotted in its posterior part; snout tip pointed laterally and extending > 1.75 mm beyond the lower jaw. Distinguished from all other species by blackish body coloration, white spots along the lateral body, and pale upper lip with dark brown spots. Character diagnostics are summarized in Table 1 and Table 2.

Description of reference specimen UADBA 19596 (RAX 3680): adult male (SVL = 39.90 mm) in excellent state of preservation. Measurements are presented in Appendix A. In the dorsal view and lateral view, the snout tip is very pointed. Snout tip with 2.30 mm straight ventral extension beyond the mouth. Head 1.52 times longer than wide. Head length 0.47 times SVL. Canthus rostralis obvious. Loreal indented. Tympanum diameter 0.80 times eye. Round tympanum with a small notch in its median superior area, clearly separated with supratympanic fold. This supratympanic fold has an umbrella-like pattern and runs to above one point between the shoulder girdle and upper arm articulation but behind one large strong granule. The anterior half part of the tympanum with light background color and the posterior part dark color. Internarial distance 0.27 times head width. Tongue ovoid anteriorly and bifid posteriorly. Non-protruding nostril with relatively close lateral aperture. Eye to nostril distance 1.59 times nostril to snout distance. Forearm length 0.47 times SVL. Hand length (including discs) 0.29 times SVL. Fingers without webbing. Outer and inner metacarpals are poorly developed. Finger relative length size 1 <2 <4 <3. Digits with a large terminal disc (the widest part twice the width of the basal disc). Tibiotarsal articulation reaching nostril. Lateral metatarsal separated. Hindlimb 1.71 times SVL. The thigh is the same length as the tibia. Foot including tarsus 0.72 times SVL. Inner metatarsal tubercle in bell-like pattern (length 1.6 mm) at base of toe 1. Outer metatarsal tubercle absent. Webbing formula 1 (0), 2i (1), 2e (0), 3i (1), 3e (0), 4i (1.5), 4e (1), 5 (0) and total sum of free phalanges 4. Relative toe length 1 <2 <3 <5 <4. Importance and repartition of body granules differently in shape and color: side and edge of the dorsal surface and above tympanum highly granulated; inguinal area, basal of flanks, and posterior of upper mouth with white evident granules; dorsum with little granules and belly finely granular. Oblong femoral glands are relatively developed with centro–distal pores surrounded by many granules, giving a crater-like pattern form. Internally, femoral glands type 3 [2,28]. In preservative, dorsum with black color. Upper lip and flanks with clean spots. The throat and the thorax with white and some silver reticulated dark brown pigments, and the belly with light yellow color. Thorax with parallel dark bands: X-like pattern on its left side and divided into two forms, spot and “+/−like” pattern on its right side. The ventral surface of the thigh is partially mottled with brown color and weak in the femoral gland. The lower part of the hindlimb is completely pigmented. The ventral side of the forelimb with clean shape and yellowish color. Evident alternate and transverse dorsal bands, shiny and dark, on the hindlimb and indistinct bands on the forelimb, but with some pinkish reticles indifferently distributed in their dorsal surfaces.

Variation: Morphometric variation is summarized in Appendix A. Sexual dimorphism is evident: males have smaller SVL (36.10–48.15 mm vs. 44.30–65.40 mm) with large eyes, a higher head, a broader terminal disc, a shorter tibia, and slightly developed nostrils. Colors vary from blackish brown (e.g., holotype, UADBA 12312, 26211, 26238, 26287, 26290, AMNH A157111, A157118, A167583, A167585) to full black one (e.g., UADBA 7769, 12313, 19647, 26240, 26242, 26298, AMNH A167588–167589). The differences are also reflected in the number of granules and pigments between these two groups of specimens: the black specimens: (1) granules almost absent and body almost smooth except on flanks, (2) with or without obvious round pigment patterns, (3) upper mouth and flank without silver and white spots, (4) ventral surface with dark brown except in belly that is a clean pattern; and the blackish brown specimens: (1) holotype with dense granules, (2) flanks with evident granules, (3) some specimens with dorsal dark spots (UADBA 12313, 26238, 26240, 26290), (4) flanks, upper mouth, throat, and thorax with obvious spots, and (5) thorax and throat with dark brown color mottled by white or silver spots and belly has no spots. The border of the tympanic region is smooth in males and finely granular in females. In males, tympanum and supratympanic fold in contact with each other except for the holotype. In addition, free phalanges on toes vary: toe 1 (0–0.50) internal edge of toe 2, (0.50–1), internal edge of toe 3 (1–1.25), external edge of toe 3 (0–0.50), internal edge of toe 4 (1–2), and external edge of toe 4 (1–1.75).

Coloration in life: The iris has a golden ring on its outer area. Just the dorsum can have a black crossband in V or Y-like pattern. The dorsum is either dark brown or thoroughly black in color. The upper mouth is either dark brown or blackish. Whitish gray spots may be present or not in the inguinal region. The ventral surface shape is very heterogeneous: throat and thorax with black color and usually with numerous small white spots; belly without pigments; and throat either with two dark spots or not. The lateral surface of the body is unicolor with no evident white spots and few granules. The ventral surface of the thigh has at least some clean pattern surfaces. Hind and forelimbs have alternate transversal bands, black and brown.

Habits: Semi-aquatic rainforest species can live in open and degraded forests, especially in Northern Madagascar. This species was observed resting along the riverbanks and streams at different stages of water speeds, between 9.00 a.m to 23.00 p.m. However, it appears rather diurnal than nocturnal and prefers rocky areas to trees. On trees, it prefers resting on leaves and branches.

Distribution: Low, mid, and high-altitude rainforest species in Central Eastern and Northern Madagascar, from 100 to 1550 m elevations (Figure 5, Appendix A).

Comments: Our morphological description agrees with the *M. mocquardi* specimen shown by [29] (Figure 1, p. 249) from Andasibe, and our molecular analyses group all our *M. mocquardi* samples with the *M. mocquardi* samples reported by [10]: AF215317 (ZFMK 66668) from Ambato, Masoala; and [11]: HQ610861 (ZSM 1846/2007) and HQ610921 (ZCMV 8818) from An’Ala and Mahasoa.

#### 3.2.5. *Mantidactylus zolitschka* [10]

Holotype: ZFMK 60110: close to An’Ala Forest (18°56′ S 48°28′ E, 840 m), 21 March 1995, F. Glaw and D. Vallan.

Paratype: ZFMK 60112–60116, ZSM 939/2000, same data as holotype. ZSM 184/2003, the same location as holotype, 2 March 2003, G. Aprea, F. Glaw, M. Puente, L. Raharivololoniaina, R. D. Randrianiaina, and M. Thomas.

Specimens examined: UADBA 6965–66, same data as holotype.

Diagnosis: A small-sized *Ochthomantis* (adult male SVL 29.6–30.6 mm, adult female SVL 37.6–37.7 mm); tibiotarsal articulation at least up to the nostril. Width of terminal disc 1.78 times basal disc. Distinguished from all other species, except *M. ambony*, by its smaller size (adult male SVL < 31 mm, female < 38 mm) and from all species by foot with less well-developed webbing (total sum of free phalanges, WS > 9). Distinguished from *M. ambony* by the absence of lateral stripe running from snout tip to inguinal region. Character diagnostics are summarized in Table 1 and Table 2.

Description of reference specimen UADBA 6966: Adult male (SVL = 27.65 mm) in good state. Measurements are presented in Appendix A. In the dorsal view, the body is clearly slender. In lateral and dorsal view, the snout tip is pointed. Snout tip with 1.20 mm ventral extension beyond the mouth. Head 1.40 times longer than wide. Head length 0.47 times SVL. Canthus rostralis are distinct and straight. Loreal is weakly concave. Tympanum diameter 0.83 times eye diameter. Round tympanum distinctly from the supratympanic fold. This tympanic fold runs straight first and rather curves in midway before reaching the forepart of the shoulder girdle and upper arm articulation. Internarial distance 0.24 times head width. Tongue ovoid anteriorly and distinctly bifid posteriorly. Small and round nostril without protuberant lateral aperture. Eye to nostril distance 1.44 times nostril to snout distance. Forearm length 0.52 times SVL. Hand length (including discs) 0.33 times SVL. Fingers without webbing. Relative finger length 1 < 2 < 4 < 3. Inner and outer metacarpal with tubercles. Digits with slightly enlarged terminal discs (widest part 1.78 times of basal disc width). Legs slender. Tibiotarsal articulation reaching nostril. Lateral metatarsal separated. Hindlimb 1.85 times SVL. Thigh the same length as the tibia. Foot including tarsus 0.78 times SVL. The inner metatarsal tubercle is rather small at the base of toe 1 (0.85 mm). Metatarsal with small tubercle. Webbing formula: 1(1), 2i (1.25), 2e (1), 3i (1.5), 3e (1), 4i (2), 4e (2), 5 (1) and total sum of free phalanges 9.75. Relative toe length 1 < 2 < 3 < 5 < 4. Skin is rather smooth on its upper surface and slightly granular on its flanks. Ventral side smooth one. Obvious femoral glands are in contact with the cloacal area and sharply delimited by granules with irregular tubercle-like patterns, and the presence of a central porus gives it a crater-like pattern. Internally, femoral gland type 3 [2,28]. In preservative, gray-brownish dorsal color with irregular dark and light marble shape. Upper lip and loreal area whitish color. Tympanic region dark brown color. Lower lip with alternate spots, light and dark. The lateral surface of the body is dark in color and the ventral surface of the body is light in color. The ventral surface of the body with different color pattern: the throat is whitish and becoming more yellowish on the belly. Throat with two longitudinal brown markings from the lip to the thorax and both merge at the shoulder girdle as a “Y-like pattern”. One light longitudinal stripe runs from the inguinal area and fades towards forelimb insertion. Light brown forelimb and hindlimb with different numerous dark crossbands (six on hand including third finger, four on thigh, three on tibia, and five on tarsus and foot). Hindlimbs with irregular dark mottling.

Variation: Morphometric variation is summarized in Appendix A. Sexual dimorphism is evident: males have smaller SVL (26.5–30.6 mm vs. 33.6–37.7 mm), and large tympanums. The free phalanges on toes vary: toe 1 (0.5–1); internal edge of toe 2 (1–1.5); external edge of toe 2 (0.5–0.75); internal edge of toe 3 (1.75–2); external edge of toe 3 (0.75–1); external edge of toe 4 (1.75–2), and in toe 5 (0.5–0.75).

Coloration in life: The iris has a golden ring on its outer area. The dorsal surface of the body has a strong gray-brownish color with a small light stripe running along it. There is a shiny yellow blotch on the inguinal region.

Habits: semi-aquatic rainforest species living close to stream, around An’Ala forest. The female ZFMK 30116 contains 49 eggs with yellow and dark brown center markings and diameter is 2 mm [10].

Distribution: Known only from type locality, An’Ala [10] (Figure 4, Appendix A).

Comments: We included genetic data for this species from [11]: HQ610866 (ZSM 1768/2007) and HQ610867 (ZSM 1841/2007).

### 3.3. Resurrected Species

Based on our molecular and morphological results, we find strong evidence to recognize two species of *Mantidactylus (Ochthomantis)* that correspond to taxa that currently are considered junior synonyms of *M. femoralis*. After examining their type specimens and our new materials, we here recognize *Mantidactylus catalai* [35] (33 specimens) and *Mantidactylus poissoni* [36] (12 specimens) as valid species and provide new descriptions for both species below (Figure 6).

#### 3.3.1. *Mantidactylus catalai* [35]

*Mantidactylus catalai* [35] (p. 203) (Holotype MNHN 1935.153, according to the original publication and [6] (p. 220); *Mantidactylus femoralis*: [22] (p. 26); *Mantidactylus (Hylobatrachus) femoralis:* [27] (p. 312); *Mantidactylus (Ochthomantis) femoralis:* [7] (p. 400), [3] (p. 3).

*M. catalai* has previously been considered by [22] as a synonym of *M. femoralis* but [10] had noted the considerable morphological differences between *M. catalai*, and *M. femoralis* of the southeast of Madagascar.

Holotype: MNHN 1935.153: Isaka–Ivondro, Tolagnaro District, Anosy Region, Madagascar, 700 m, 1935, M. R. Catala. Specimen in good condition.

Specimens examined: AMNH 7881–7882 A133689–133690, AMNH 18019 A168364: Fianarantsoa–Ifanadiana Road, Southwest Ranomafana, Ifanadiana District, Vatovavy–Fitovinany Region, Madagascar, 900 m. AMNH A 181732*, A 181821* (RAX 10563, 10599): Beampingaratsy Pass, Anosy Montain, Tolagnaro District, Anosy Region, Madagascar, (24°28.244′ S 46°53.521′ E, 490–1140 m), 12 February–13 February 2009, S. Mahaviasy, N. Rakotondrazafy, and C. J. Raxworthy. UADBA 1419–1421, 1423 (RAN 36377, 36434, 36446, 36505): Ampasimekieny Pass, Tolagnaro District, Anosy Region, Madagascar (24°32.0′ S 46°51.0′ E, 800–950 m), 24 December–28 December 1990, J. B. Ramanamanjato, A. Raselimanana; C. J. Raxworthy, A. Razafimanantsoa, and A. Razafimanantsoa. UADBA 3706 (RAN 35091): Manatantely Forest, Tolagnaro District, Anosy Region, Madagascar (29°59.0′ S 46°55.083′ E, 125 m), 30 October 1990, same collectors as UADBA 1419. UADBA 4513–4514; 4516, 4521, 4523 (RAN 52831, 52699, 52762, 52807, 52830): Eminiminy, Andohahela National Park, Tolagnaro District, Anosy Region, Madagascar (24°35.04′ S 46°44.08′ E, 1000–1100 m), 11 November–15 November 1995, J. B. Ramanamanjato, and A. Raselimanana. UADBA 4522 (RAN 52472): Ambinany, Andohahela National Park, Tolagnaro District, Anosy Region, Madagascar (24°35.6′ S/46°44.3′ E, 820 m), 14 November 1995, J. B. Ramanamanjato, and A. Raselimanana. UADBA 9772–9774; 9782 (RAN 57002, 56723, 56937, 57006): Amorimbato Forest; Kalambatritra Special Reserve, Iakora District, South–Est Region, Madagascar (23°27.44′ S 46°20.02′ E, 1150–1300 m), 30 October–8 November 1996, J B Ramanamanjato, R. A. Nussbaum, and J. Spannring. UADBA 26403–26405 (RAN 44835, 44672, 44701): Sahavatoy and Volontsagana Rivers, Andringitra National Park, Ihorombe Region, Madagascar (22°13.667′ S 47°0.217′ E, 810–1240 m), 24 November–30 November 1993, N. Rabibisoa, A. Razafimanantsoa, and C. J. Raxworthy. UMMZ 191515–191516 (RAN 32567, 32597): Sainte Luce, Tolagnaro District, Anosy Region, Madagascar (24°45′ S 47°11′ E, 20 m), 7 October and 10 October 1989, R. A. Nussbaum, and C. J. Raxworthy. UMMZ 197662–197664 (RAN 35686, 35706–35707): Nahampoana, Tolagnaro District, Anosy Region, Madagascar (24°58′ S 46°58′ E, 75–300 m), 23 November–24 November 1990, R. A. Nussbaum, J. B. Ramanamanjato, and A. Raselimanana. UMMZ 197676 (RAN 36626): Manangotry, Tolagnaro District, Anosy Region, Madagascar (24°45′ S 46°52′ E, 850 m), 3 January 1991, J. B. Ramanamanjato, A. Raselimanana, and C. J. Raxworthy. UMMZ 212890* (RAN 44491): Iatara River, Andringitra National Park, Ivohibe District, Atsimo Atsinanana Region, Madagascar (22°13.333′ S 47°01.483′ E, 720 m), 18 November 1993, same collectors as UADBA 26403.

Diagnosis: A medium to large-sized *Ochthomantis* (adult male SVL 41–45 mm, female SVL 51–62 mm); tibiotarsal articulation between eyes and nostril (or very rarely at snout); toes fully webbed, except on toe 4 where 1–1.5 phalanges are free; no stripe line along superior lip; inguinal region with clean pattern area; snout tip very pointed in lateral view with large extension beyond mouth (1.75–3.45 mm); head wider and flattened but very sharp as “fish-like” pattern. Distinguished from other species by the following characters: *M. ambreensis* and *M. ambony* by the absence of white stripes along the lateral body; *M. femoralis*, *M. zolitschka*, and *M. danieli* n. sp. by the number of free phalanges in the internal edge of toe 4 (1–1.5) and absence of prominent pale yellow or white stripes in the inguinal region (horizontal or oblique); *M. mocquardi* by the number of free phalanges on the internal edge of toe 4 (1–1.5) and flanks without whitish spots; *M. olgae* n. sp. by the absence of obvious black granules on flanks and absence of crossbars in V- or Y-like pattern on dorsum in preservative; *M. tavaratra* n. sp. by digits with large terminal discs (widest part > 1.80 times of basal disc width), lack of prominent and pale inguinal streak, and absence of white strip on superior lip; *M. poissoni* by the absence of white spots below the eye and tibiotarsal articulation between eyes and nostrils; *M. macrotympanum* n. sp. by smaller adult male SVL (<60 mm). Character diagnostics are summarized in Table 3 and Table 4.

Description of reference specimen UADBA 1419 (RAN 36377): Adult male (SVL = 42.60 mm) in good state of preservation. Measurements presented in Appendix A. In the dorsal and lateral view, the snout tip is pointed. Snout tip with 2.60 mm ventral extension beyond the mouth. Head 1.40 times longer than wide. Head length 0.50 times SVL. Canthus rostralis well distinct. Loreal region with evident indentation. Tympanum diameter 1.03 times eye. Slightly round tympanum in contact with supratympanic fold, except in its posterior part, and this tympanic fold runs behind tympanum towards the small granule above shoulder girdle and forearm articulation. Tympanum with a small notch in its median superior area. Internarial distance 0.27 times head width. Tongue ovoid anteriorly and bifid posteriorly. Nostrils with distinct cutaneous fold and lateral oblique aperture. Eye to nostril distance 1.58 times nostril to snout distance. Forearm length 0.61 times SVL. Hand length (including discs) 0.42 times SVL. Fingers without webbing. Inner and outer metacarpal tubercles are evident. Relative fingers length 1 < 2 < 4 < 3. Digits with large terminal discs (widest part 2.07 times of basal disc width). Tibiotarsal articulation between the eye–nostril. Lateral metatarsal separated. Hindlimb 1.84 times SVL. The thigh is the same length as the tibia. Foot including tarsus 0.78 times SVL. The inner metatarsal tubercle is obvious along toe 1 (2.15 mm). The outer metatarsal tubercle is small and granule-shaped. Webbing formula: 1 (0.75) 2i (1) 2e (0.25), 3i (1.25) 3e (0.25) 4i (1.5) 4e (1) 5 (0) and total sum of free phalanges 6. Relative toe length 1 < 2 < 3 < 5 < 4. Body granules vary: flank with obvious granules, dorsum with granules irregularly distributed, granules of tympanum more concentrated above its superior part, and sacral area and belly finely granulated. Oblong femoral glands are relatively developed and its medio-proximal part with pores surrounded by some granules, giving it a crater-like pattern. Internally, femoral gland type 3 [2,28]. In preservative, the dorsal surface of the body is brown in color and the white longitudinal vertebral band runs from the snout tip to the anal pore. Lips with two vertical light stripes on the loreal region. Shiny line behind the eyes. The ventral surface of the body is orange–brown in color but has a clean pattern. The throat has a couple of dark brown spots (in front of the shoulder girdle). The inguinal region with white spots or not. The ventral surface of the forearm is bordered by a brown-sided but clean pattern. Fore and hindlimbs with obvious alternate crossbands, dark and shiny.

Variation: Morphometric variation is summarized in Appendix A. Sexual dimorphism is evident: males have smaller SVL than females (41.10–45.40 mm versus 51.85–61.50 mm), wider head, bigger tympanum, longer fore and hindlimbs, and metatarsal tubercle not evident. The ratio of td/ed in males is larger than in females (0.89–1.08 vs. 0.50–0.69).

The femoral glands are more swollen in males and smaller and more circular in females. A majority of all specimens are brown darker color, except for the holotype and UADBAs (1419–1421, 4513, 9773, 26403–26404) which are light brown In the dorsal view, granules in a plate-like pattern are observed except for UADBAs (4523, 26403, 26405). A vertebral line is absent, except for UADBAs (1419, 4513, 4523). The throat with white spots in males, except for UDBAs (1419, 1421). The free phalanges of toes vary: toe 1 (0–1), internal edge of toe 2 (1–1.25), external edge of toe 2 (0–0.25), internal edge of toe 3 (1–1.25), external edge of toe 3 (0–0.50), internal edge of toe 4 (1–2), external edge of toe 4 (1–1.50), and in toe 5 (0–0.25).

Coloration in life: The dorsal surface of the body is dark brown. Superior lips with light dots. Round tympanum with central dark color surrounded by dark background. Male without yellowish vertebral line. Inguinal region with small spots or not. Reddish brown ventral surfaces with small spots and darker punctuations.

Habits: semi-aquatic rainforest species living in bamboo forests. This species is diurnal and/or nocturnal, observed between 9.00 a.m to 0.15 a.m, which is adapted to a “burrowing life”, inside of holes and interstice of rocks, but close to the waters (stagnant water to the river but very rarely in a fast stream). It is mainly observed on rocks and sometimes on the ground, especially during the day. No individuals were seen on leaves or branches, but one individual UADBA 4521 was collected on roots.

Distribution: Known only from rainforest in Southeast Madagascar at low and medium elevations (Figure 7, Appendix A):

Comments: Our morphological description agrees with the *M.* sp. aff. *mocquardi* shown by [29] (Figure 3, p. 249) from Ambatolahy, near Ranomafana, and our molecular analyses group all our *M. catalai* samples (Appendix A) with the Ranomafana *M.* cf. *mocquardi* sample reported by [11]: AY324821 (FGMV 2002.173). This specimen has more recently been referred to as “Confirmed Candidate Species (CCS) sp. 47” [11]. The LSID number is E9176C8E-291B-48BD-9857-BF3D2FEF1517.

#### 3.3.2. *Mantidactylus poissoni* [36]

*Mantidactylus poissoni* [36] (p. 178) (Holotype MNHN 1937.1) *Mantidactylus femoralis*: [22] (p. 26); *Mantidactylus (Hylobatrachus) femoralis:* [27] (p. 312); *Mantidactylus femoralis*: [37]: (p. 278); *Mantidactylus (Ochthomantis) femoralis:* [7] (p. 400), [3] (p. 3).

Holotype: MNHN 1937.1: Mandraka forest, 70 km from Antananarivo, Manjakandriana District, Analamanga Region, Madagascar, 1937, collected by M. H. Poisson. This type is in a poor state of preservation according to our observation and [22] (p. 45).

Reference specimen: AMNH A174653* (RAX 9367): Mandraka, Manjakandriana District, Analamanga Region, Madagascar, (18°54.727′ S 47°55.174′ E, 1250 m), 18 March 2006, N. Rabibisoa, J. Rafanomezantsoa, N. A. Rakotondrazafy, and P. Razafimahatratra. Same locality as the holotype.

Specimens examined: MNHN 1937.1: Mandraka Forest, 70 km from Antananarivo, Manjakandriana District, Analamanga Region, Madagascar, 1937, M. H. Poisson. AMNH A50362 adult male: Madagascar, 1971, Guibé. AMNH A174649-50 (RAX 8198–8199) adult females: Manasamena River, Lakato, Moramanga District, Alaotra Mangoro Region, Madagascar (19°02’38.2″ S 48°20’54.6″ E 950 m), 29 March 2004, N. Rabibisoa, M. Randriambahiniarime, and F. Ranjanaharisoa. AMNH A174653* (RAX 9367): Mandraka, Manjakandriana District, Analamanga Region, Madagascar (18°54.727′ S 47°55.174′ E, 1250 m), 18 March 2006, N. Rabibisoa, J. Rafanomezantsoa, N. A. Rakotondrazafy, and P. Razafimahatratra. UADBA 6876, 7125 adult females: Ankeniheny and Andasibe, Moramanga District, Alaotra Mangoro Region, Madagascar (19°05.850′ S 48°19.910′ E, 950 m, and 18°57′ S 48°26′ S, 900 m), 28 December 1994 and 15 December 1997, N. Rabibisoa, and S. Ramilison. UADBA 11899 (NR 1196) adult male: Sahaberiana, Mantadia National Park, Moramanga District, AlaotraMangoro Region, Madagascar (18°47,503′ S 48°25,572′ E, 895 m), Nov. 20, 1998, J. Rafanomezantsoa, and N. Rabibisoa. UADBA 19786 (LV77) subadult female: Ambatovaky Special Reserve, Soanierana Ivongo District, Analanjirofo Region, Madagascar (16°46.910′ S 49°14.417′ E, 600 m), 5 August 1999, by N. Rabibisoa, and S. Ramilison. UADBA 26409 (RAN 45665) adult female: Rangovalo Ridge, Zahamena National Park, Fenoarivo Atsinanana District, Analanjirofo Region, Madagascar (17°40.5′ S 48°45.5′ E, 1150 m), Mar. 4, 1994, J. B. Ramanamanjato, A. Raselimanana, C. J. Raxworthy, and A. Razafimanantsoa. UADBA 26411–26412 (RAX 8190, 8155) adult females: Manasamena River, Lakato, Moramanga District, Alaotra Mangoro Region, Madagascar (19°02.637′ S 48°20.910′ E 950 m), Mar. 29 and 27, 2004, N. Rabibisoa, M. Randriambahiniarime, and F. Ranjanaharisoa. UADBA 39000 (RAX 9368) adult male: same data as a reference specimen.

Diagnosis: A medium to large-sized *Ochthomantis* (adult male SVL 39.7–48.2 mm, adult female 53–66 mm); tibiotarsal articulation reaching at least nostril; width of digit terminal disc ≥ 1.70; one large white spot underneath of eye for females and numerous white spots for males. Distinguished from all other species by the following combination of characters: *M. ambreensis* and *M. ambony* by lack of white or yellow continuous line on lateral surfaces of head and body; *M. femoralis* by lack of prominent and pale inguinal patch or line and upper lip without white stripe; *M. mocquardi* by lack of white spots along lateral surface of body and surfaces of body neither black nor very dark brown color; *M. catalai* by the presence of white spots below eye and tibiotarsal articulation reaching at least nostril; *M*. *olgae* n. sp. by the absence of black granules on dorsal surface of head and lateral surface of body; *M. tavaratra* n. sp. and *M*. *danieli* n. sp. by presence of white spots below eye; *M. zolitschka* by its larger size (SVL ≥ 39.7 mm); and *M. macrotympanum* n. sp. by its smaller adult male size (SVL < 49 mm). Character diagnostics in Table 3 and Table 4.

Description of UADBA 39000 (RAX 9368): adult male (SVL = 48 mm) in excellent state of preservation. Measurements are presented in Appendix A. In the dorsal view and lateral view, the snout tip is relatively obtuse. Snout tip with 1.35 mm ventral extension beyond the mouth. Head 1.25 times longer than wide. Head length 0.45 times SVL. Canthus rostralis distinct. Loreal with groove. Tympanum diameter 0.98 times eye. Slightly round tympanum with a vivid small notch in the middle of the superior area and in contact with the supratympanic fold in its anterior part and separate in its posterior part, and this supratympanic fold runs towards before reaching the shoulder girdle and upper arm articulation. Internarial distance 0.25 times head width. Tongue ovoid anteriorly and bifid posteriorly. Round nostrils with distinct cutaneous fold and lateral aperture. Eye to nostril distance 1.81 times nostril to snout distance. Forearm length 0.49 times SVL. Hand length (including discs) 0.28 times SVL. Fingers without webbing. Inner and outer metacarpals exist. Fingers without webbing. Relative finger length 1 < 2 < 4 < 3. Digits with large terminal discs (widest part > 1.70 times of basal disc width). Tibiotarsal articulation between nostril and snout tip. Lateral metatarsal separated. Hindlimb 1.76 times SVL. The thigh is the same length as the tibia. Foot including tarsus 0.72 times SVL. The inner metatarsal tubercle is shield-shaped (length 2 mm) at the base of the toe 1. Outer metatarsal tubercle in the small granule. Webbing formula: 1 (0), 2i (1), 2e (0), 3i (1.25), 3e (0.50), 4i (1.75), 4e (1.75), 5 (0.25) and total sum of free phalanges 6.50. Relative toe length is 1 < 2 < 3 < 5 < 4. Lateral surfaces of the body and dorsum with numerous granules. Little and swollen femoral glands elongated, and its pore on the media-distal area was surrounded by many granules, giving it a crater-like pattern. Internally, femoral glands type 3 [2,28]. In preservative, the dorsal surface of the body with blackish brown color. Upper lip with shiny transverse and interrupted band oriented to eyes. The belly and ventral surface of the thigh with yellowish-white spots. Obvious white pigments on boundary surfaces of the thorax and abdomen and making them together an 8-like pattern. Thorax and throat were almost pigmented by whites with some scattered brown spots and the thorax with two brownish parallel dark bands. The inguinal region with whitish L-shaped bed pattern. Dorsal transverse bands are rather indistinct on the hind and forelimbs.

Variation: Morphometric variation is summarized in Appendix A. Sexual dimorphism is evident: males have a smaller SVL (39.7–46.8 mm), thicker snout, shorter hand, larger terminal disc, and loreal less elongated. The ratio of td/ed is larger in males than in females (0.62–0.82 vs. 0.51–0.68). Tibiotarsal articulation between nostril and snout tip except UADBAs (19786, 26411, 26412) and AMNH A174650 beyond snout tip. Femoral glands are more swollen in males and smaller in females. Free phalanges on toes vary: toe 1 (0–0.75), external edge of toe 2 (0–0.25), internal edge of toe 3 (1–1.50), external edge of the toe 3 (0–0.25), internal edge of the toe 4 (1.25–1.75), external edge of the toe 4 (1–1.50), and in toe 5 (0–0.25).

Coloration in life: The iris has a golden ring on its outer area with some black spots. The dorsal surface of the body with a dark brown color has more small granules. There is a more or less round yellow spots in the inguinal region for females and a stick-like pattern for males. The ventral surface of the body is pigmented by a white color with yellowish border bands in an 8-like pattern. Hind and forelimbs with alternate dark and shiny brown transverse bands.

Habits: Semi-aquatic rainforest species living close to small streams with rock. Stream depth does not reach 1 m. All specimens were observed during the day and night between 16.00 pm to 21.30 pm. In the daytime, they were observed between 5–10 m far from the bank on the ground and at night 3–5 m far from the river, and roosting on leaves between 50–100 cm in height. UADBA 7125 was seen on rocks in the middle of the river where water is very speedy, maybe ending up there by misfortune. It is rather a ground and tree-dwelling than an aquatic frog.

Distribution: Mid and high elevations of the eastern slope forest of Madagascar (Figure 6, Appendix A).

Comments: The only genetic data known for this species are the sequence from the reference specimen AMNH A174653 (RAX 9367) (Appendix A). The LSID number is 57C53062-B2DF-41E5-88A3-EFE7B083DA84.

### 3.4. New Species Descriptions

Based on our molecular and morphological results, we find strong evidence to recognize four species of *Mantidactylus (Ochthomantis)* that correspond to taxa that cannot be assigned to any nominal species (or to names considered as junior synonyms). After examining our new materials, and developing diagnoses for each taxon, we here provide descriptions for each of these new species, such as *Mantidactylus danieli* n. sp. (54 specimens), *M. macrotympanum* n. sp. (5 specimens), *M. olgae* n. sp. (60 specimens), and *M. tavaratra* n. sp. (150 specimens) (Figure 8).

#### 3.4.1. *Mantidactylus danieli*, New Species

Holotype: AMNH A167590* (RAX 4268) adult female collected 22 February 2002 at Salafaina Forest, District Vohemar, Sava Region, Madagascar, 400 m, 13°26.257′ S 49°43.001′ E by S. Mahaviasy and N. Rabibisoa.

Paratypes: AMNH A167523* (RAX 6595) juvenile collected 2 April 2003 at the relict Irony Forest, Antsohihy District, Sofia Region, Madagascar, 950 m, 14°45.140′ S 48°29.690′ E by S. Mahaviasy, N. Rabibisoa, N. Rakotondrazafy and C. J. Raxworthy; AMNH A167582 (RAX 2999) adult female collected 9 April 2001 at Ramena River, Ambanja District, Diana Region, Madagascar, 750 m, 13°55.071′ S 48°53.179′ E, by S. Mahaviasy, N. Rabibisoa, C. J. Raxworthy, A Razafimanantsoa, and A. Razafimanantsoa; AMNH A 167592 (RAX 4371) adult female, AMNH A 167591 (RAX 4339) and UADBA 19595 (RAX 4372) adult males collected 24–25 February 2002, the same locality as the holotype; AMNH A 167517 (RAX 6110) and UADBA 26359 (RAX 6118) adult males, collected 11 March 2003 at Matsaborimena Trois Lacs, 1550 m, 14°19.859′ S 48°35.240′ E, Bealanana District, Sofia Region by S. Mahaviasy, N. Rakotondrazafy, and N. Rabibisoa; AMNH A181773* (RAX 10204) collected 5 April 2008 at Andramanalana, Andapa district, Sava Region, Madagascar, 850 m, 14°22.351′ S 49° 21.747′ by S. Mahaviasy and N. Rakotondrazafy; AMNH A181731* (RAX 10392) collected 16 April 2008, Tsararano, Anjanaharibe–Sud/Masoala corridor, Analanjirofo Region, Madagascar, 490 m, 14°54.667′ S 49°41.383′ E, by S. Mahaviasy and N. Rakotondrazafy; UADBA 3716 (RAN 39507) adult male, collected 9 March 1992 at Bekolosy Manongarivo, Ambanja District, Diana Region, Madagascar, 1200 m, 14°02.5′ S 48°18′ E by J-B. Ramanamanjato, A. Raselimanana, and C. J. Raxworthy; UADBA 7770 (MRJ 108) juvenile, collected 12 October 1992 at Ampanasatongotra, Marojezy National Park, Andapa District, Sava Region, Madagascar, 600 m, 14°26.2′ S 49°46.5′ E by N. Rabibisoa, J. B. Ramanamanjato, and O. Ramilison; UADBA 8382 (RAX 2737) adult female, collected 9 April 2001: at Ramena River Analabe, Tsaratanana Reserve, Ambanja District, 750 m, 13°55.071′ S 48°53.179′ E by S. Mahaviasy, N. Rabibisoa, C. J. Raxworthy, A Razafimanantsoa, and A. Razafimanantsoa; UADBA 19593 (RAX 3454) adult female, collected 4 December 2006 at Ambolokopatrika, Anjanaharibe–Sud/Marojejy Corridor, Andapa District, Madagascar, 880 m, 14°32.302′ S 49°26.243′ E, the same collectors as holotype; UADBA 19594 (RAX 3785), adult male collected 15 December 2001 at Andranomavohely, Andapa District, Sava Region, Madagascar, 800 m, 14°34.165′ S 49°16.568′ E, by S. Mahaviasy, N. Rabibisoa, N. Rakotondrazafy, A. Razafimanantsoa, and A. Razafimanantsoa; UADBA 26361 (RAX 6482) adult female, collected 21 March 2003, Analapakila, Trois Lacs, 1450 m, 14°26.233′ S 48°36.696′ E, District Bealanana, Sofia Region by S. Mahaviasy, N. Rakotondrazafy and N. Rabibisoa; UADBA 26366–26368, 26373 (RD 918, 839–840, 881), collected November 2000 at Ambolokopatrika, 880 m, 14°32.302′ S 49°26.243′ E, Andapa, Sava Region by D. Rakotomalala; UADBA 26371 (RAX 3783) adult male, the same condition as the holotype; UMMZ 212827* (RAN 38186) collected 22 November 1991 at Antomboka River, Montagne d’Ambre, Antsiranana, Diana Region, 1150 m, 12°32.3′ S 49°10′ S by C J Raxworthy, J B Ramanamanjato, and A Raselimanana; UMMZ 212835 (RAN 39291) adult male: collected 24 February 1992 at Ambalafary, Manongarivo Special Reserve, Ambilobe District, Diana Region, Madagascar, 250 m, 14°04′ S 48°17′ E, by C. J. Raxworthy, A. Raselimanana, J. B. Ramanamanjato, A. Razafimanantsoa, and A. Razafimanantsoa; UMMZ 212836* (RAN 39387) collected 2 March 1992, Antsahabe River, Manongarivo Special Reserve, Antsiranana, Diana Region, 1200 m, 14°02.5′ S 48°18′ E by C. J. Raxworthy, J. B. Ramanamanjato, and A. Raselimanana.

Additional specimens examined: See Appendix C.

Diagnosis: A small to medium-sized *Mantidactylus (Ochthomantis)* species (adult SVL male 33–42 mm, female 42–59 mm), with dark brown color on the dorsal surface of the body and often with black spotted granules. The inguinal region with a soft pale yellow streak and may be partly broken up or narrow; no continuous pale stripe along the lower flank; lack of obvious white spot below the eye; moderately developed foot webbing with 1.5–2 free phalanges at the internal edge of toe 4 and WS > 7; tibiotarsal articulation extends beyond nostril; and maximum width of a terminal disc on fingers < 1.70 disc width base. *Mantidactylus danieli* can be distinguished from the following species: *M. mocquardi*, *M. catalai*, *M. olgae* n. sp., and *M. tavaratra* n. sp. by more developed webbing: WS > 7, 1.5–2 free phalanges at the internal edge of toe 4, tibiotarsal articulation beyond nostril, and presence of pale yellow inguinal streak that may be partly broken up or narrow; *M. poissoni* by lack of white spots below eye; *M. femoralis* by a maximum width of the terminal disc on fingers < 1.70 base width and by inguinal pale yellow streak may be partly broken up and narrow; *M. zolitschka* by larger adult SVL > 32 mm; *M. ambreensis* and *M. ambony* by absence of continuous pale stripe along lower flank; and *M. macrotympanum* n. sp. by smaller adult SVL (<60 mm) and smaller male tympanum/eye diameter < 0.94. Table 5 and Table 6 summarize the characters’ diagnosis for this new species.

Description of holotype: Adult female (SVL 50.6 mm) in excellent state of preservation. Measurements are presented in Appendix A. In the dorsal view, the head is longer than the width. Head length 1.18 times width length. Head length 0.41 times SVL. In dorsal view, the snout tip is pointed and rounded lateral, with a 3 mm ventral extension beyond the lower lip. Canthus rostralis evident. Loreal concave. The tympanum and supratympanic fold are distinct from each other, and this supratympanic fold runs towards the front of the upper arm and shoulder girdle articulation. Round tympanum diameter 0.74 times eye diameter. Tongue ovoid anteriorly and bifid posteriorly. Internarial distance 0.21 head width. Round nostril with lateral aperture. Eye–nostril distance 1.27 times nostril–snout distance. Forearm length 0.51 times SVL. Hand length (including discs) 0.26 times SVL. Fingers without webbing. Outer and inner metacarpal with developed tubercles. Relative finger length 1 < 2 <4 < 3.

The terminal is disc relatively large (the widest part is 1.45 width of its basal disc). Tibiotarsal articulation between nostril and snout tip. Hindlimb 2.01 times SVL. Thigh length 0.98 times tibia length. Foot including tarsus 0.92 times SVL. Lateral metatarsal separated. Inner metatarsal tubercle as a skin-like pattern with three-sided (1.35 mm) on base of toe 1. Outer metatarsal tubercle present. Webbing formula: 1 (1), 2i (1), 2 (0.25), 3i (1.75), 3e (0.50), 4i (2), 4e (2), 5 (0.50), and total sum of free phalanges is 9. Relative toe length 1 < 2 < 3 < 5 < 4. The dorsal surface of the thigh, the posterior area above the eye, above the tympanum, and around the femoral glands with granule skin-like patterns. Round femoral glands poorly developed with central pores. Internally, femoral gland type 3 [2], but without small granules in the proximal area, characteristics of male femoral glands [28]. In the preservative, the dorsal surface of the body is dark brown in color and has small darker spots. Upper lip dark brown color except under tympanic region clean pattern. The throat is pale brown in color with two dark parallel bands in an L-shape. The belly is pale brown in color and is almost entirely covered with fine dark brown spots, which decrease in density posteriorly. Brown forelimbs without dark bands. Brown hindlimb with darker transversal bands on its dorsal surface and pale brown with dark-brown spots on its ventral surface. The inguinal region with pale and discontinuous oblique streak.

Variation: Morphometric variation is summarized in Appendix A. Sexual dimorphism is obvious in this species: males are smaller sized with relatively larger tympanum than females, smaller SVL than females (33–42 mm vs. 42–59 mm), larger tympanum diameter than females (0.66–1.12 times eye diameter vs. 0.49–0.76 times eye diameter), shorter hindlimb, and larger terminal disc. Free phalanges on the toes vary: toe 1 (0.50–1), internal edge of toe 2 (0.50–1.50), external edge of toe 2 (0–0.75), internal edge of toe 3 (1–1.75), external edge of toe 3 (0–1), internal edge of toe 4 (1.75–2), external edge of toe 4 (1.50–2), and in toe 5 (0–0.50).

Coloration in life: The iris color is golden in the superior area and fades to the lower area of the eye. The dorsum and dorsal surfaces of the head and limbs have a dark brown color and sometimes black spots for females. The upper lip has a white band that is most developed in the posterior area. Between the eyes, there is a darker brown cross-bars in a **V**-like pattern. The ventral surfaces with dark brown spot pigments have a different background color: the belly is creamy or yellowish but the paler pattern is almost dark reddish on its anterior part, two dark reddish parallel stripes may be present on the throat and become narrower in its posterior part and decline on the thorax or is absent (UADBA 8382, 26361, 26366, 26367, 26371, 26373). The inner area of the thigh is completely covered with dark reddish-brown spots. A pale yellow oblique streak in the inguinal region is typically discontinuous with all of the specimens, except for UADBA 19595, 26359, and 26367. Transversal bands, black and darker brown were observed on the dorsal surface of the hindlimb, but indistinctly clear with some dark specimens (e.g., UADBA 26406). One adult female (UADBA 3770) has a pale vertebral line running from the snout to the anal pore.

Habits: Living in rainforests between 350–1580 m elevation and semi-aquatic species; occupying the banks of small streams and rivers. Observed between 10.30–23.00 h, although more active at night and appears to be mostly nocturnal. Unlike other *Ochthomantis* species, males rest mostly on river banks, more rarely on leaves of trees and bushes, and very rarely on rocks. Females are more tree-dwelling than rupicolous. The vertical distribution appears to be different between sexes: males between 20–100 cm, whereas females are between 50–300 cm above the ground. This species appears to prefer rivers to small fast-flowing streams and females were observed mostly in areas of slow-moving water.

Etymology: The specific name danieli is a patronym, i.e., a noun in the genitive case honoring Daniel Rakontondravony for his substantial contributions to the knowledge of the Malagasy fauna.

Distribution: Low- and mid-altitude Morthern and Sambirano Region of Madagascar (180–1200 m), from subhumid to humid forests (Figure 9, Appendix A).

Comments: Our morphological description agrees with the *M.* sp. aff. *femoralis* specimen shown by [29] (pp. 178–179) from Tsaratanana. Our molecular analyses group all our *M. danieli* samples (Appendix A) with previously published sequences from Montagne d’Ambre AY324818 (FGMV 2002.929) and Manongarivo AY324816 (FGMV 2002.825). *Mantidactylus* cf. *femoralis* samples reported by [10]. FGMV 2002.929 has more recently been referred to as “CCS sp. 42”. The LSID number is 3A4FC339-EB42-4262-8B61-0F26A27B7505.

#### 3.4.2. *Mantidactylus macrotympanum* New Species

Holotype: AMNH A 167589* (RAX 2715): adult male at Antsahatelo, Western Slope Tsaratanana Strict Reserve, Ambanja District, Diana Region, Madagascar, 800 m, 13°51.588′ S 48°51.979′ E, 6 April 2001, N. Harilanto, S. Mahaviasy, N. Rabibisoa, C. J. Raxworthy, A. Razafimanantsoa, and A. Razafimanantsoa.

Paratypes: AMNH 167598 (RAX 2603): adult male at Antsahabe, Ramena River, Ambanja District, Diana Region, Madagascar, 180 m, 13°43.272′ S 48°39.304′ E, 4 April 2001, N. Harilanto, S. Mahaviasy, N. Rabibisoa, C. J. Raxworthy, A. Razafimanantsoa, and A. Razafimanantsoa. UMMZ 201416*–201418 (RAN 39170–39171, 39470), UMMZ 213447* (RAN 39125): adult males, Ambalafary, Manongarivo Special Reserve, Ambilobe District, Diana Region, Madagascar, 250 m, 14°04′ S 48°17′ E, 20 February 1992 and 3 March 1992, A. Raselimanana, J. B. Ramanamanjato, C. J. Raxworthy, A. Razafimanantsoa, and A. Razafimanantsoa.

Diagnosis: A large-sized *Mantidactylus (Ochthomantis)* species (adult male SVL 57–62 mm, female unknown but presumably like other *Ochthomantis* species, female larger than males > 62 mm); dorsal surfaces of head and body with dark brown color; body with skin granules especially on its posterior parts; no pale inguinal streak; large tympanum ≥ 7.6 mm; tibiotarsal articulation beyond snout tip; and webbing well-developed and total sum of free phalanges 3.75. *Mantidactylus macrotympanum* can be distinguished from all species in the subgenus *Ochthomantis* by the following characteristics: big tympanum diameter ≥ 7.6 mm, a large adult male SVL ≥ 57 mm, and presence of skin granules on lateral and dorsal surfaces of the body. Table 5 and Table 6 summarize the characters’ diagnosis for this new species.

Description of holotype: Adult male (SVL 61 mm) in excellent state of preservation. Measurements are presented in Appendix A. In dorsal view, the head is longer than wide: head length 1.29 times head width. Head length 0.44 times SVL 0.44. In the dorsal view, the snout tip is pointed, and in the lateral view almost acuminate and curved down with 2.20 mm ventral extension beyond the lower lip. Obvious canthus rostralis and concave loreal making lip well evident. Tympanum and supratympanic fold distinct and supratympanic fold running vertically in its posterior part to one evident granule above the upper arm and shoulder girdle articulation. Round tympanum diameter 0.94 times eye diameter. Tongue ovoid anteriorly and bifid posteriorly. Internarial distance 0.23 times head width. Non-protruding nostril with lateral slanting aperture. Eye–nostril distance 1.86 times nostril–snout tip distance. Forearm length 0.50 times SVL. Hand length (including discs) 0.32 times SVL. Fingers without webbing. Outer and inner metacarpal tubercles flattened and widened. Relative finger lengths 1 < 2 <4 <3. Terminal discs are large (the widest part is 2.20 times of basal disc width). Tibiotarsal articulation largely beyond the snout tip. Hindlimb 1.89 times SVL. Thigh and tibia same length. Foot including tarsus 0.76 times SVL. Lateral metatarsal separated. Outer metatarsal tubercle absent. The inner metatarsal tubercle is not evident at the base of toe 1. Webbing formula 1 (0), 2i (0.5), 2e (0), 3i (1), 3e (0), 4i (1.25), 4e (1), 5 (0) and total sum of free phalanges 3.75. Relative toe length 1 < 2 < 3 < 5 < 4. Flanks with few prominent granules in its superior part and a white spot-like pattern posteriorly. Numerous epidermic granules in the sacral area and finely granules above the eyes. Round femoral glands are not swollen. Internally, femoral gland type 4 [2,28]. In preservative, the dorsal surface of the body is blackish and marbled with some shiny spots. Ventral surfaces of the body and superior lip homogenous pattern with a brownish color. The throat is darker with two parallel bands. Fore and hindlimbs with alternate transversal bands, shiny and dark. The inguinal region has some scattered dark spots.

Variation: Morphometric variation is summarized in Appendix A. Species are only known from six male specimens. SVL between 57.9 and 62.1 mm. The ventral surface of the body is heterogeneous according character of the belly: individuals with a clean pattern belly have black spots and those can be either present or not with dark belly specimens. Flanks are shiny brownish backgrounds with white plates in the central area like the holotype, but sometimes white spots are observed for specimens that have black flank backgrounds. Three internal phalanges vary between 0.50–1 for toe 2i and 1–1.50 for toe 4i.

Coloration in life: The iris has a constant black color surrounded by white color in its superior and lower parts. The body is dark brownish. The superior lip has a black crossbar anteriorly. Above the eyes, there is a white line. The belly is darker with a black spot. Thorax has two parallel bands. The vivid brown flanks have a white plate-like. A white oblique line can be present in the inguinal area. Two alternate bands, shiny and dark, are rather distinct on the fore and hindlimbs.

Habits: This is a rainforest species living along calm streams and fast rivers with scattered rocks. Individuals were also found outside the forest, e.g., in plantations at Manongarivo. Individuals were observed between 12.00–20.00 h but they are more nocturnal than diurnal. At night, they roost and rest on branches between 10–200 cm in height and during the day, they rest on the rocks. All the known specimens observed are males.

Etymology: The specific name macrotympanum is composed of the Latin words macro (large) and tympanum (tympanum), referring to the very big tympanum of this species. The name is used as a noun in apposition.

Distribution: Known only from relict forest close to Ambalafary village, the Manongarivo Special Reserve at 250 m, and from the Ramena River (Tsaratanana Reserve) between 180–800 m (Figure 10, Appendix A).

Comments: The only genetic data known for this species were obtained from AMNH A167589 (RAX 2715), UMMZ 201416 (RAN 39170), and UMMZ 213447 (RAN 39125) (Appendix A). The LSID number is 957B5BBF-17E1-4175-A499-2953919D7F89.

#### 3.4.3. *Mantidactylus olgae*, New Species

Holotype: AMNH A167596* (RAX 5195), subadult female collected 19 April 2002 at Sorata pic Forest, Vohemar District, Sava Region, Madagascar, 1700 m, 13°41.147′ S 49°26.511′ E, S. Mahaviasy, N. Rakotondrazafy, N. Rabibisoa, and C.J. Raxworthy.

Paratypes: AMNH A167555 (RAX 2475) adult male, and UADBA 8383, 8388 (RAX 2473, 2472) adult females collected 25 January 2001 at Besahona, Analabe forest, Ambanja District, Diana Region, Madagascar, 700 m, 13°54.372′ S 48°44.785′ E, A. Razafimanantsoa, and A. Razafimanantsoa; AMNH A167556 (RAX 2552) adult female collected 29 January 2001 at Ramena, Analabe forest, Ambanja District, Diana Region, Madagascar, 700 m, 13°42,707′ S 48°34,156′ E, same collectors as previous. AMNH A167557 (RAX 2555) adult male collected 29 January 2001 at Betaindambo, Analabe forest, Ambanja District, Diana Region, Madagascar, 600 m, 13°51.932′ S 48°49.189′ E, same collectors as previous; AMNH A167558 (RAX 2620) adult female collected 4 April 2001 at Ramena, Analabe forest, Ambanja District, Diana Region, Madagascar, 700 m, 13°42.707′ S 48°34.156′ E, N. Harilanto, S. Mahaviasy, N. Rabibisoa, C.J. Raxworthy, A. Razafimanantsoa, and A. Razafimanantsoa; AMNH A167560 (RAX 2655) adult male, and AMNH A167561–167562 (RAX 2662–63) adult females collected 5 April 2001 at Maroamalona, Analabe forest Ambanja District, Diana Region, Madagascar, 600 m, 13°51.023′ S 48°47,902′ E, same collectors as previous. AMNH A167565* (RAX 3203) juvenile, and AMNH A167567–167568 (RAX 3230, 3232) adult males, collected 15–16 April 2001 at Antsaravy Analabe forest, Ambanja District, Diana Region, Madagascar, 1150 m, 13°55.560′ S 48°54.353′ E, N. Harilanto, S. Mahaviasy, N. Rabibisoa, A. Razafimanantsoa, and A. Razafimanantsoa; AMNH A181726* (RAX 10205) collected 5 April 2008 at Andramanalana, Andapa District, Sava Region, Madagascar, 850 m, 14°24′ 24″ S 49°20’13″ E, S. Mahaviasy, and N. Rakotondrazafy; UADBA 8231, 8387, 8389–8390 (RAX 3231, 3228, 3196–3197) adult females, collected 15–16 April 2001 at Antsaravy Analabe forest, Ambanja District, Diana Region, Madagascar, 1150 m, 13°55.560′ S 48°54.353′ E, N. Harilanto, S. Mahaviasy, N. Rabibisoa, C.J. Raxworthy, A. Razafimanantsoa, and A. Razafimanantsoa; UADBA 8404 (RAX 3312) adult male, collected 19 April 2001 at Antsaravy, Analabe forest, Ambanja District, Diana Region, Madagascar,1150 m, 13°55.560′ S 48°54.353′ E, N. Harilanto, S. Mahaviasy, N. Rabibisoa, A. Razafimanantsoa, and A. Razafimanantsoa.

Diagnosis: A small- to medium-sized *Mantidactylus (Ochthomantis)* species (adult SVL male SVL 34–40 mm, female SVL 45–52 mm) with gray color of the body; dorsal and lateral surfaces of the body with black granules; no pale yellow streak in the inguinal region; no broad pale line or white spots on the lateral surface of the body; upper lip with a pale line; the presence of crossbars in V or Y–like pattern on dorsum; snout tip pointed in lateral view; well-developed foot webbing with 1 free phalange on internal edge of toe 4 (WS < 7); tibiotarsal articulation between eye and nostril; and maximum width of terminal disc on fingers > 1.8 times of basal disc width. Distinguished from all others species by presence of obvious black skin granules both on dorsal and lateral surfaces of body and by the following characters: *M. femoralis* by one free phalange on internal edge of toe 4, no yellow patch on inguinal region, and tibiotarsal articulation between eye and nostril; *M. ambreensis* and *M. ambony* by lacking of broad pale line on lateral surface of body; *M. mocquardi* by gray body color and no white spots on lateral surface of body; *M. zolitschka* by larger male adult SVL > 33 mm and female adult > 44 mm; *M. poissoni* by lack of white spots below eye; *M. catalai* by presence of crossbars in V or Y-like pattern on dorsum; *M. danieli* n. sp. by following characters:webbing more developed (WS < 7), one free phalange at internal edge of toe 4, no pale yellow streak in inguinal region, and tibiotarsal articulation between eye and nostril; *M. macrotympanum* n. sp. by smaller male adult SVL < 60 mm and smaller male tympanum eye diameter < 0.94; and *M. tavaratra* n. sp. by larger terminal disc and maximum width of terminal discs on finger > 1.8 times of basal disc width. Character diagnostics are summarized in Table 5 and Table 6.

Description of holotype: Subadult female (SVL = 40.70 mm) in excellent state of preservation. Measurements are presented in Appendix A. In the dorsal view, the head is rather longer than the width: the head length is 1.23 times the head width. Head length 0.46 times SVL. In dorsal and lateral view, the snout tip is pointed with evident ventral extension 2.20 mm beyond the lower lip giving it a shell-like pattern. Canthus rostralis not evident. Loreal furrowed. The tympanum was distinct from a well-developed suratympanic fold which ran and reached towards one point before upper arm and shoulder girdle articulation. Tympanum diameter 0.56 times eye diameter. Tongue ovoid anteriorly and bifid posteriorly. Internarial distance 0.21 times head width. Round nostrils with cutaneous fold anterolateral aperture. Eye–nostril distance 1.74 times nostril–snout tip distance. Forearm length 0.55 times SVL. Hand length (including discs) 0.33 times SVL. Fingers without webbing. Outer and inner metacarpal tubercles are poorly developed. Relative fingers length 1 < 2 < 4 < 3. Larger terminal disc, i.e., widest part twice of basal disc width). Tibiotarsal articulation between eye and nostril. Hindlimb 1.98 times SVL. Thigh length 1.07 times tibia length. Foot including tarsus 0.77 times SVL. Lateral metatarsal separated. Inner metatarsal tubercle in half moon-like pattern (1.3 mm length) at base of the toe 1. Outer metatarsal with small granule. Webbing formula: 1 (0), 2i (1), 2e (0), 3i (1), 3e (0), 4i (1.25), 4e (1), 5 (0) and total sum of free phalanges 4.25. Relative toe length 1 < 2 < 3 < 5 < 4. Skin granules on the flank. Round femoral gland poorly developed with a central pore. Internally, femoral gland type 3 [2] but without small granules in its proximal area characteristics of the male femoral gland [28]. In preservatives, coloration is grayish dorsally with a Y-like pattern. Upper lip with white pigments. Throat with two dark parallel bands. Belly, ventral surface of the thigh, and forelimb clean patterns. Hind and forelimbs with gray color and alternate dorsal bands, shiny and dark, crossing them. An inguinal region without a pale inguinal streak. Flank with some black granules.

Variation: Morphometric variation is summarized in Appendix A. Sexual dimorphism is evident: males have a smaller SVL than females (34.00–39.55 mm vs. 45.75–51.80 mm), males have a larger tympanum diameter than females (0.80–1.30 times eye diameter in males vs. 0.40–0.69 times eye diameter in females), and larger hindlimb; 90% of specimens have a larger thigh, between 0.91–1.18 times tibia length and the rest < 0.91; free phalanges vary: toe 1 (0–0.50), internal edge of toe 3 (1–1.25), external edge of toe 3 (0–0.50), and internal edge of toe 4 (1–1.50).

Coloration in life: The iris has a golden ring on its outer area. Body coloration is gray except for brown color, AMNH A167560 and UADBA (8391, 8477). The upper lip is white and dirty except for the holotype which is white in color. No dark parallel bands on the throat except for UADBA 8407. All of the specimens have V or Y–like patterns with dark spots on the dorsum except UADBA (8384, 8387, 8389) with X–like patterns. The ventral surface of the body is clean pattern except for juveniles which are dirty and some specimens have a throat with white pigments including the holotype and AMNH (A167557, A167564, A167568–167569, A167571, A167573–167574). The inguinal region has no pale yellow streak. The dorsal hindlimbs have darker brown transversal bands. Some specimens have a thick pale brown vertebral line.

Habits: Low- and mid-altitude rainforest species close to the edge of streams or rivers with rocky bottoms. We observed it between 10.00–22.30 h and mainly nocturnal except for juveniles. It was observed in various habitats: rock, grounds, riverbanks, and edges of water. Generally, females rest on rocks at night and some specimens overhanging on leaves between 20–50 cm above the ground. Males are more tree-dwelling than rupicola and rest on leaves of shrubs, between 10–30 cm in height. Specimens observed at all types of water, but they have a particular preference for streams or fast rivers.

Etymology: The specific name olgae honors the late Olga Ramilijaona, professor at UADBA and supervisor of the Ph.D. thesis of N.R., in recognition of her substantial mentorship.

Distribution: Species inhabits along Ramena River (Tsaratanana Strict Natural Reserve, and Analabe), Andramanalana, and Sorata in the North of Madagascar. The elevation ranges between 600–1700 m (Figure 11, Appendix A).

Comments: The only genetic data known for this species are presented in Appendix A. Although *M. olgae* has been recorded at Andramanalana, no genetic or morphological samples have yet been confirmed from Marojejy, which is a well-sampled site about 40 km to the west. The LSID number is FB2EEFBB-92C7-460E-892A-9B5BBFF65862.

#### 3.4.4. *Mantidactylus tavaratra*, New Species

Holotype: AMNH A 167505* (RAX 5310) adult male collected 22 April 2002 at Sorata, Vohemar District, Sava Region, Madagascar, 980 m, 13°41.986′ S 49°26.687′ E, by S. Mahaviasy, N. Rabibisoa, and C. J. Raxworthy.

Paratypes: AMNH A 167506* (RAX 5400) juvenile, AMNH A167507 (RAX 5403) adult female, and UADBA 26306 (RAX 5599) adult male collected 3 February 2003 at Matsaborimaiky Lake, Tsarananana Strict Natural Reserve, Bealanana, Sofia Region, Madagascar, 1950 m, 14°09.175′ S 48°57.431′ E, by S. Mahaviasy, N. Rabibisoa, N. Rakotondrazafy, and C. J. Raxworthy. AMNH A167510* (RAX 5769) juvenile collected 23 February 2003 at Befosa River, Tsaratanana Strict Natural Reserve, Bealanana District, Sofia Region, Madagascar, 1680 m, 14°10.455′ S 48°56.708′ E, by S. Mahaviasy, N. Rabibisoa, N. Rakotondrazafy, and C. J. Raxworthy. AMNH A167515*–167516* (RAX 6028, 6107) adult males collected 11 March 2003 at Matsaborimena, Trois Lacs, Bealanana District, Sofia Region, Madagascar,1600 m, 14°19.859′ S 48°35.240′ E, by S. Mahaviasy, N. Rabibisoa, and N. Rakotondrazafy. AMNH 167524*–167525* (RAX 6838, 6901) adult males, and AMNH A167529 (RAX 7120), UADBA 26326, 26335, 26394 (RAX 7009, 6903, 6952) adult females collected 13–17 April 2003 at Lohanandroranga River ridge, and Ambatotavaratra River, Corridor Tsaratanana Anjanaharibe–Sud, Bealanana District, Sofia Region, Madagascar, 1430–1650 m, 14°24.076′ S 49°10.253′ E, the same collectors as AMNH A167510. AMNH A167593* (RAX 5000) adult male collected 13 April, 2002 at Sorata, Vohemar District, Sava Region, Madagascar,1330 m, 13°41.147′ S 49°26.511′ E, by S. Mahaviasy, N. Rabibisoa, and C. J. Raxworthy. AMNH A181730* (RAX 10323) juvenile 10 April 2008 at Andramanalana, Andapa, Sava Region, Madagascar, 1300 m, 14°24.400′ S 49°20.21′ E, by S. Mahaviasy, and N. Rakotondrazafy. AMNH A187088*–187089* (RAX 11534–11535), and UADBA RAX 11532*–11533* juveniles collected 6 May 2010 at Marojejy, Andapa District, Sava Region, Madagascar, 1550 m, 14°32.302′ S 49°26.243′ E, by S. Mahaviasy, N. Rakotondrazafy, and C.J. Raxworthy. UADBA 19628, 19634 (RAX 4993, 4999): adult females, and UADBA 19630 (RAX 4995) adult male collected 13 April 2002 at Sorata, Vohemar District, Sava Region, Madagascar, 1300 m, 13°41.147′ S 49°26.511′ E, by S. Mahaviasy, N. Rabibisoa, and C. J. Raxworthy. UADBA 19625, 19631, 26307–26308 (RAX 4951, 4996, 5771, 5778) adult males, and UADBA 26328, 26331 (RAX 5402, 5753) adult females collected 22–23 February 2003 at Befosa River, Tsaratanana Strict Reserve, Bealanana District, Sofia Region, Madagascar, 1600 m, 14°10.455′ S 48°56.708′ E, by S. Mahaviasy, N. Rabibisoa, and N. Rakotondrazafy. UADBA 19650 (RAX 3980) adult female collected 11 February 2002 at Bezavona, Vohémar District, Sava Region, Madagascar, 350 m, 13°31.962′ S 49°51.954′ E, by S. Mahaviasy, N. Rabibisoa, and C. J. Raxworthy. UMMZ 212440* (RAN 39071) juvenile collected 16 February 1992 at Bekolosy, Manongarivo Special Reserve, Ambanja District, Diana Region, Madagascar, 1100 m, 14°02.5′ S 48°18′ E, by A Raselimanana, J. B. Ramanamanjato, and C. J. Raxworthy. UMMZ 212889* (RAN 43366) adult female collected 1 April 1993 at Befosa River, Tsaratanana Natural Reserve, Bealanana District, Sofia Region, Madagascar, 1630 m, 14°10.455′ S 48°56.708′ E, by J. B. Ramanamanjato, A. Raselimanana, C. J. Raxworthy, A. Razafimanantsoa, and A. Razafimanantsoa.

Additional specimens examined: See Appendix C.

Diagnosis: A medium to large-sized *Mantidactylus (Ochthomantis)* species (adult SVL male 35–52 mm, female 42–63 mm); brown dorsum without black granules; lateral and ventral surfaces of body well-distinct by different background color, from darker to pale pattern respectively; pale white or yellow thin streak in the inguinal region; upper lip with pale stripe; webbing with 1–1.5 free phalanges at external edge of toe 4 and no free phalange at the external edge of toe 2; tibiotarsal articulation between eye and nostril; and maximum terminal disc width of fingers ≤ 1.8 basal disc width. Distinguished from all other species by the following characteristics: *M. ambreensis* and *M. ambony* by lack of evident and white or yellow continuous line on lateral surfaces of head and body; *M. poissoni* by lack of white spots below eye; *M. catalai* by digits with smaller terminal discs (widest part ≤ 1.8 of basal disc width), thin pale white or yellow oblique line in inguinal region and white stripe on upper lip; *M. femoralis* and *M. danieli* n. sp. by the presence of 1–1.5 free phalanges at the external edge of toe 4 and tibiotarsal articulation beyond nostril; *M. mocquardi* by the presence of thin pale continuous line on the upper lip, absence of white spots on the flank, and absence of gray body coloration; *M. olgae* n. sp. by flank typically with straight border from dark to pale venter coloration and maximum width of terminal disc ≤ 1.80 basal disc width; *M. zolitschka* by larger adult SVL > 35 mm; and *M. macrotympanum* n. sp. by smaller male adult SVL < 60 mm and smaller male tympanum-eye diameter < 0.94. Character diagnostics in Table 5 and Table 6 for this new species

Description of holotype: Adult male (SVL = 37 mm) in excellent state of preservation. Measurements are presented in Appendix A. In the dorsal view, the head is larger than the width: the head length is 1.34 times the head width. Head length 0.44 times SVL. In the dorsal view, the snout tip is slightly pointed and almost straight in the lateral view with a 1.55 mm ventral extension beyond the lower lip, which makes the snout tip have an obtuse appearance. Canthus rostralis obvious and loreal concave making the jaw well evident. Tympanum and supratympanic fold distinct running almost vertically towards three evident white granules in front of upper arm and shoulder girdle articulation. Round tympanum diameter 0.62 times eye diameter. Tongue ovoid anteriorly and bifid posteriorly. Internarial distance 0.30 times head width. Round nostril with non-protruding lateral aperture. Eye–nostril distance equals nostril–snout distance. Forearm length 0.46 times SVL. Hand length (including discs) 0.30 times SVL. Fingers without webbing. Outer and inner metacarpal tubercles flattened and widened. Relative finger length 1 < 2 < 4 < 3. Fingers without webbing. The terminal disc is relatively large (the widest part is 1.40 times the basal disc width). Tibiotarsal articulation between eye and nostril. Ventral surface of the forearm with three granules on its internal ridge at the base of metacarpal tubercles. Hindlimb 1.90 times SVL. The thigh length I almost equal to the tibia length. Foot including tarsus 0.77 times SVL. Lateral metatarsal separated. Outer metatarsal tubercle indistinct. Inner metatarsal tubercle flattened at base of toe 1. The webbing formula is: 1 (0.25), 2i (1), 2e (0), 3i (1.25), 3e (0.25), 4i (1.5), 4e (1.5), 5 (0), and the total sum of the free phalanges is 5.75. Relative toe length 1 < 2 < 3 < 5 < 4. The dorsal surface of the body with small granules on its edge and on the sacral region. Oblong femoral glands are relatively developed and in their centro-distal area with pores surrounded by numerous granules, giving a crater-like pattern. Internally, femoral gland type 3 [2,28]. In preservative, brown dorsum with some clean pattern or depigmented portions, especially in its posterior region. Upper lip with finely dark brown pigments posteriorly and covering with obvious round pigments anteriorly. The indistinct band runs behind the eyes and crumbles in the anterior half of the dorsum. Throat with two dark short parallel stripes. Ventral surfaces are homogeneous for different parts of the structure: throat and thorax with silver–white pigments and some reticulated dark brown pigments, belly whitish in color with some brown round spots, thigh and forearm no pigments in its part of the area. The inguinal region has fine streaks. The lower portion of the hindlimb with few scattered brown spots. Hindlimb with evident alternate bands, thick dark, and fine shiny. Forelimb with indistinct alternate bands.

Variation: Morphometric variation is summarized in Appendix A. Sexual dimorphism is obvious in this species: males have smaller SVL than females (35.65–52.40 mm vs. 42.70–63.25 mm), males have a larger tympanum diameter than females (0.82–1.12 times eye diameter in males vs. 0.43–0.73 times eye diameter in females). Body coloration varies from grayish brown to reddish brown UADBA 26331 (RAX 5753) or light brown UADBA 19628, 26308 (RAX 4993, 5778). In males, half of the specimens (including the holotype) have snout tips ≤ 2 mm ventral extension beyond the lower lip and the other half > 2 mm. Throat with obvious dark brown parallel bands, except for some specimens with the ventral surface of the body and clean pattern: e.g., UADBA (26238, 26371, 26394). Throat and thorax with white or white silver coloration, except for UADBA (26238, 26308, 26331, and 26394). Dark dorsal crossbar in V or Y-like pattern absent, except for holotype and paratypes UADBA (19625, 19628, 19630–19631, 26308, 26394). Flanks with obvious black and white granules were observed on paratype UADBA 26391 but did not reach the roughness of *M. olgae* n. sp. Toe 3 is slightly shorter compared to toe 5 in males, whereas it is shorter in females, except UADBA 26326 (RAX 7009). Free phalanges vary: toe 1 (0–0.75), external edge of toe 2 (0–0.75), internal edge of toe 3 (0.25–1.50), external edge of toe 3 (0–0.50), internal edge of toe 4 (1–2), and external edge of toe 4 (1–1.50).

Coloration in life: The iris has a golden ring on its outer area. The body is grayish brown, light, dark brown, or sometimes reddish brown. The upper lip has an evident vivid white band or not. Between the eyes, there is a darker brown cross-bars in a rod-like pattern. The dorsal surface of the body has dark brown cross-bars in a V or Y-like pattern. The throat and thorax are silvery white or white in color. The throat is always with distinct dark brown parallel stripes. The belly has a clean pattern with smaller white spots. The inner side of the thighs has some dark reticulations. The inguinal area has a white or yellowish little line or not, but if it exists the line is narrower in shape. White brownish bands can be present on the outer side of the forelimbs and alternate transversal bands, white and brown on the inner side of the forelimbs. The dorsal surface of the hindlimbs has alternate transversal bands, light and dark browns. The black granules may be present or not on the white flank but concentrated in the posterior part of the dorsum and on the side of the forearm, but they are not very obvious like *M. olgae*.

Habits: This is a rainforest and savannah species living along the riverbank. It inhabits almost all kinds of aquatic habitats: streams, rivers, ponds, and shallow marshes except waterfalls. We observed it between 9.30–23.00 h, but mainly nocturnal except juveniles. Juveniles observed during the day. At night, they rest both on rocks and leaves, but rarely on branches and accidentally on aerial roots and *Pandanus* sp. The females prefer roosting on rocks rather than on leaves, whereas the males are indifferent to those. The vertical distribution varies between the sexes: males between 5–100 cm and females between 5–150 cm above the ground.

Etymology: The specific name tavaratra refers to the Malagasy word for “north”. This name is used as a nonlatinized noun in apposition and is given in reference to the known distribution of this species in Northern Madagascar

Distribution: Northern endemic species of Madagascar, elevation varies from 80 to 2450 m elevation (Figure 12, Appendix A).

Comments: Our morphological description agrees with the *M.* sp. aff. *mocquardi* shown by [29] (Figure 2, p. 249) from Marojejy. Our molecular analyses group all our *M. tavaratra* samples (Appendix A) with those Manongarivo AY324819 (FGMV 2002.824) and Tsaratanana AY324820 (ZSM 643/2001), reported as *M.* cf. *mocquardi* by [10]. FGMV 2002.824 from Manongarivo has more recently been referred to as “UCS sp. 63 Tsaratanana”, and this taxon also includes another specimen (FGMV 2001.114) from Tsaratanana [11]. The LSID number is BD3F2F68-88CF-4629-9A91-0196E2CF2F4A.

### 3.5. Identification Key

Lateral head and body with a broad pale line running continuously from the snout tip to the groin area. *M. ambreensis* (SVL, adult male between 33–39 mm and adult female 38–42 mm), and *M. ambony* (SVL, adult male between 30–32 mm and adult female 34–38 mm).
-Lateral head and body without a broad pale line running continuously from the snout tip to the groin area. 2.Adult male SVL < 31 mm, female SVL < 38 mm, reduced foot webbing with ≥2 free phalanges on the internal edge of toe 4 and WS > 9, male tympanum diameter ≤ 7 mm. *M. zolitschka*.
-Adult male SVL 31–55 mm, female SVL 38–67 mm, more extensive foot webbing with ≤2 free phalanges on the internal edge of toe 4 and WS < 9, male tympanum diameter < 7 mm. 3-Adult male SVL > 56 mm (female unknown, but likely ≥ 67 mm), more extensive foot webbing with 1.25 free phalanges on the internal edge of toe 4 and WS < 4, male tympanum diameter ≥ 7 mm. *M. macrotympanum*.Tibiotarsal articulation reaches between nostril and snout tip, or beyond. 4.
-Tibiotarsal articulation reaches between eye and nostril. 6.Upper lip with a large pale spot under the eye (female) or multiple white spots (male), groin area lacks a short pale line or spots. *M. poissoni*.
-Upper lip lacks a large pale spot under the eye (female) or multiple white spots (male), groin area with a short pale line or spots. 5.Large finger pads with disc width > 1.7 × disc base, upper lip with a thin pale continuous line, groin area with a short bold pale line that may be continuous or broken (in life, yellow). *M. femoralis*.
-Smaller finger pads with disc width < 1.7 × disc base, upper lip without a thin pale continuous line, groin area with a short weak pale line that is often broken or even absent. *M. danieli*.Upper lip with a thin pale continuous line. 7.
-Upper lip without a thin pale continuous line but may have small pale spots. 8.Snout tip sharply angular in lateral view, flanks with an irregular border from the dark to pale venter coloration, dorsal surface of body gray in life (and often in preservation) with black granules, larger finger pads, with disc width > 1.8 × disc base. *M. olgae*.
-Snout tip not sharply angular in lateral view, flanks typically with a straight border from the dark to pale venter coloration, dorsal surface of body brown without black granules, smaller finger pads, with disc width ≤ 1.8 × disc base. *M. tavaratra*.Snout tip angular in lateral view, body dark brown (or black in preservation) with distinct white spots on flanks, groin area with a few or lacking pale spots, foot webbing with 1.5 free phalanges on the internal edge of toe 4 and WS 4. *M. mocquardi*.
-Snout tip not angular shape in lateral view, body medium brown (never black in preservation) and lacks white spots on flanks, groin area with a prominent short pale line or spots, foot webbing with 1–1.5 free phalanges on the internal edge of toe 4 and WS 6. *M. catalai*.

## 4. Discussion

### 4.1. Taxonomy

According to this study, the subgenus *Ochthomantis* contains 11 species—five species previously considered to be valid, *M. femoralis* [8], *M. mocquardi* [32], *M. ambreensis* [30], *Mantidactylus ambony* [5], and *M. zolitschka* [10]; two resurrected species, *M. catalai* [35] and *M. poissoni* [36]; and four newly described species, *M. olgae* n. sp., *M. danieli* n. sp., *M. tavaratra* n. sp., and *M. macrotympanum* n. sp.

The increased species number is in agreement with the previous study focusing on tadpoles [11] which identified numerous candidates for species in the subgenus, several of which were validated by our study with adult specimens through morphological and cladistic analysis [28].

One species currently included in the subgenus *Ochthomantis*, *M. majori* (Figure 13), should not be included in this subgenus based on unpublished Principal Component Analysis, which places the species completely separate from the other species of the subgenus *Ochthomantis*, and on our cladistic analysis on 96 morphological patterns, habitats, and behavioral characters [28].

The divergence of *M. majori* is evidenced by its different reproductive behavior (parental care, time, and mating behavior). Based on our field observation, we confirm that males of *M. majori* guard eggs and practice parental care (Figure 13a) as [38,39] observed too; probably to protect eggs against fungi [40], predators, and desiccation. Egg deposition takes place on leaves overhanging the water (Figure 13b) in contrast to other *Ochthomantis* species where it takes place at edges of riverbanks either on twigs or on rocks, as we observed in *M. femoralis* and *M. ambreensis* (Figure 7d and Figure 13c). Furthermore, morphological differences in *M. majori* to other *Ochthomantis* are very evident, i.e., it has very developed webbing (toes almost fully webbed, WS = 0–2.5), hindlimbs very short, internal space more developed than interorbital space, presence of the thin line vertebral, toe 3 slightly shorter than toe 5, presence of the thin white line in the side of hindlimbs and especially inguinal region free of obvious pigments (white or yellow spots; line or band patterns). However, our molecular analyses recover *M. majori* as a sister to *M. macrotympanum*, and this new subclade forms a paraphyletic group with the remnant of the subgenus *Ochthomantis* species with moderate support (posterior probability = 0.91); thus, while we do not include *M. majori* in this revision, we recommend that the placement of *M. majori* is further investigated in a study with broader taxonomic and genetic sampling.

We confirm *M. flavicrus* as a junior synonym of *M. femoralis*. The analysis of the type of *M. flavicrus*—from the description of [8], the measurement from [10], and an examination made by CJR of the type specimen of *M. flavicrus* based on our identification key—supported that they both represent the same species.

We noticed that two species in our analysis are composed of genetic subclades, *Mantidactylus mocquardi* and *M. ambreensis* (Figure 1); however, the genetic distances between the subclades are small (<3%) but may be further explored in future studies. The first of these species, *M. mocquardi*, has two subclades representative of populations from the north and south of the Sambirano River, respectively. The north of the Sambirano specimens are characterized by their black color, while those from the south of the Sambirano have a more brownish color. These subgroups also differ by additional characters: for the totally blackish specimens, (1) granules almost missing, body smoothing and homogeneous, (2) white spots or silvery in flanks and missing on superior lips, and (3) ventral face marbled with brown reticulate and white pigment except on belly that is a clean pattern in preservative; for the brownish specimens (1) granules evident on flanks, (2) white pigment very conspicuous on lower flanks continuing to lower lips, (3) some dark spots observed on back with different shape: UADBA 12313 (NR 1372), UADBA 26290 (RAX 8036), UADBA 26238 (RAX 8021), UADBA 26240 (RAN 45476), (4) ventral face completely white in living specimen, while thorax and throat with brown spots and marbled with white or silvery pigments in contrasting of clean pattern belly-shaped in preservative. Additional specimens and observation of *M. ambreensis* especially from southern populations, Ramena, and Irony, are needed to better discern morphological differences including body size.

### 4.2. Sexual Dimorphism and Size

Within the subgenus *Ochthomantis*, sexual dimorphism is evident in SVL and tympanum size. Males have a larger tympanum than females (0.62–1.33 vs. 0.39–0.79). The tympanum senses sound waves and detects the position and direction of animal movement [41]. Here, sexual dimorphism in tympanum size may be related to reproductive communication. The tympanum in males can play a role not only for receiving but also for emitting sound by vibration, which especially makes sense given their small vocal sacs, where their call has low intensity [6,35]. The tympanum in females only functions as a receiver, thus its smaller size. This may suggest that attraction by calls over long distances is less significant for *Ochthomantis* compared to other anuran groups that breed during the warm and rainy season in Madagascar.

*Ochthomantis* females are characterized by a larger body size than males. This characteristic is a mechanism to reduce the effort spent during egg laying [42] and allows for an increase in the number and size of eggs which in some dissected individuals occupied almost the whole of the ventral cavity [28]. The smaller size of males can be also explained by: 1) a higher energy spending due to their higher activity during reproduction and competition with other males.

Some other morphometric characters may be related to the ecology of each species: (1) longer hindlimbs, and thus longer tibias may be related to improved jumping capabilities and are found in species with more terrestrial behavior (*M. femoralis*, *and M. danieli*) than in species more specialized to living on rocks and in aquatic environments (*M. olgae, M. mocquardi, M. tavaratra*, and *M. catalai*), (2) *M. catalai* has pointed snout and flattened head which may be an adaptation to move within in rock fissures, (3) Terminal discs of phalanges are probably used to cling and hang on leaves or rocks. Accordingly, the more tree-dwelling and rock-dwelling species (e.g., *M. olgae, M. tavaratra*) have a larger terminal disc than those most ground-dwelling species such as *M. danieli*, (4) well-developed webbing is observed especially in the more aquatic and rock-dwelling species (*M. olgae, M. mocquardi, M. tavaratra*) compared to the terrestrial and arboreal-dwelling species (*M. femoralis, M. danieli*). Webbing is used for propulsion in the water and increases the surface to grip the rocks, 5). Generally, species with stocky bodies often have long hindlimbs. In addition, according to our observations, they appear to be more dynamic jumpers (e.g., *M. femoralis*, *M. danieli*).

### 4.3. Period of Reproduction and Reproduction

The observations made in the north and the region of Moramanga indicate that the reproduction at least of some species of *Ochthomantis* is spread throughout the year. Thus, the statement from [6] that reproduction happens during the cool season is partially true for the species of low altitude and mid-altitude in the central eastern region. The beginning of reproduction probably takes place in April during which the temperature falls to 10 °C and metamorphosis of tadpoles occurs between October and November. While for species from the high altitude in the North (>1400 m asl) as *M. tavaratra*, the reproductive season starts very early when the temperature falls to 8 °C, e.g., at Tsaratanana the tadpoles in advanced stages were observed at the end of February (stage 36). It could be hypothesized that in mid and high altitudes, the activating factor may be the decline of temperature (around 10 °C in Moramanga, and 8 °C in Tsaratanana at high altitude, 2300 m elev. asl). We also noticed that the metamorphosis from the larval stage to the juvenile stages appears to occur fast at high altitudes. This speed may be necessary to avoid predation or other factors [43], e.g., the drying up of streams at high altitudes for instance on the Tsaratanana massif.

### 4.4. Biogeography, Climate, and Conservation

We noticed that *Ochthomantis* are absent in the West and South of the Island. We propose three hypotheses related to their reproduction habits. The first deals with temperatures, as the factor activating mating in species of the subgenus, appears to be a decline of temperature (between 8–15 °C), and it is clear that these regions of the island have a high temperature all year long. The second is the lack of water during the dry winter season when most of the streams are dried up. Finally, the third possible factor could be the absence of consistently calm water for breeding given that during the summer season, the water velocity can be very quick after torrential rains.

Global climate change has the potential to influence the fate of *Ochthomantis* as high mountain species have difficulty rapidly adapting to new conditions or migrating to areas with suitable conditions [44]. For *Ochthomantis*, climate change may also disturb the period of reproduction, particularly in those species where a decrease in temperature is critical to the induction of reproduction. In addition, changes in water regimes in the high mountains may cause the disappearance of breeding sites due to desiccation, or to a reduction in calm stretches of water with an increased frequency of torrential rains.

Data presented during a dedicated workshop held in Madagascar on 28 January 2009 [45] projected warming across the island, with Southern Madagascar being most affected and the coast and the north showing lower projected temperature increases. Precipitation increase was projected to be centered in the northwest while drying was projected in the East.

Integrating climate change into conservation strategies for amphibians, and specifically for the subgenus *Ochthomantis*, requires more precise distribution model forecasts under alternative climate scenarios, and we anticipate that amphibians occurring at high elevations will show higher sensitivities than those from low elevations [46]. For ground-truthing these scenarios, long-term monitoring efforts are also needed [47].

## 5. Conclusions

This study resolves the taxonomy of various cryptic species in the subgenus *Ochthomantis* using both morphological characters and molecular data. Our analysis revealed four new species that are formally named and described herein: *M. danieli* n. sp, *M. macrotympanum* n. sp, *M. olgae* n. sp, *M. tavaratra* n. sp. Furthermore, *Mantidactylus catalai* and *M. poissoni* are revalidated from junior synonyms to good species. The subgenus *Ochthomantis* now contains 11 species, but we suspect that some specimens included here may represent additional distinct species. It is hoped that the discovery and taxonomic revision of this new cryptic biodiversity will initiate conservation activities for those species with the most restricted distributions.

## Figures and Tables

**Figure 1 animals-13-02800-f001:**
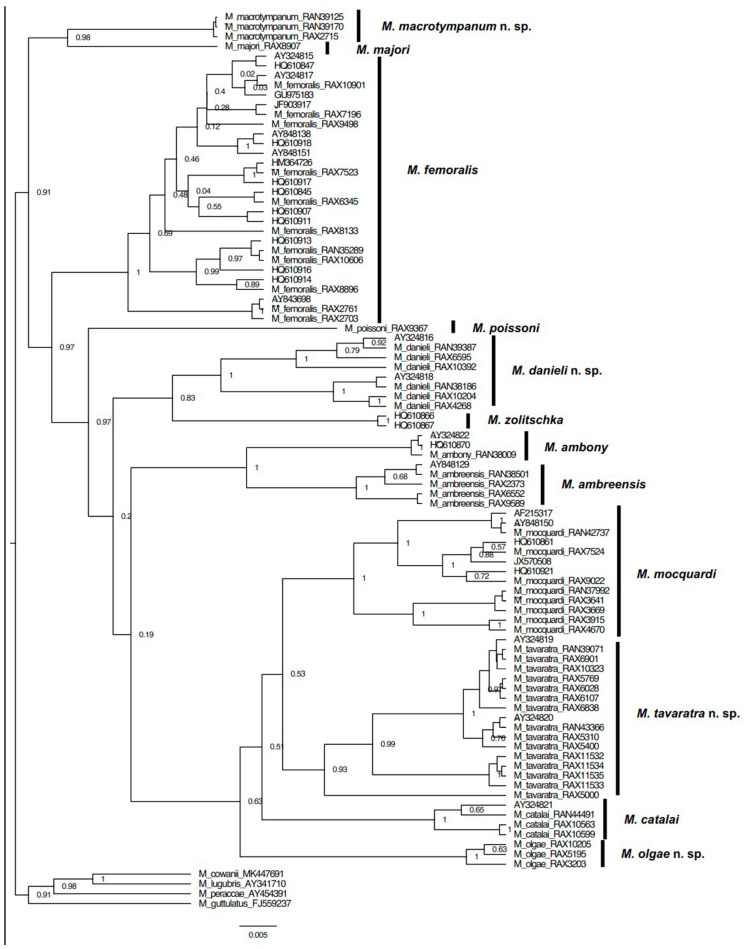
16S rRNA gene tree of samples of the subgenus *Ochthomantis* from a BEAST analysis. Posterior probabilities support molecular species groupings. *Mantidactylus cowani*, *M. lugubris*, *M. peraccae*, and *M. guttulatus* were used as outgroup.

**Figure 2 animals-13-02800-f002:**
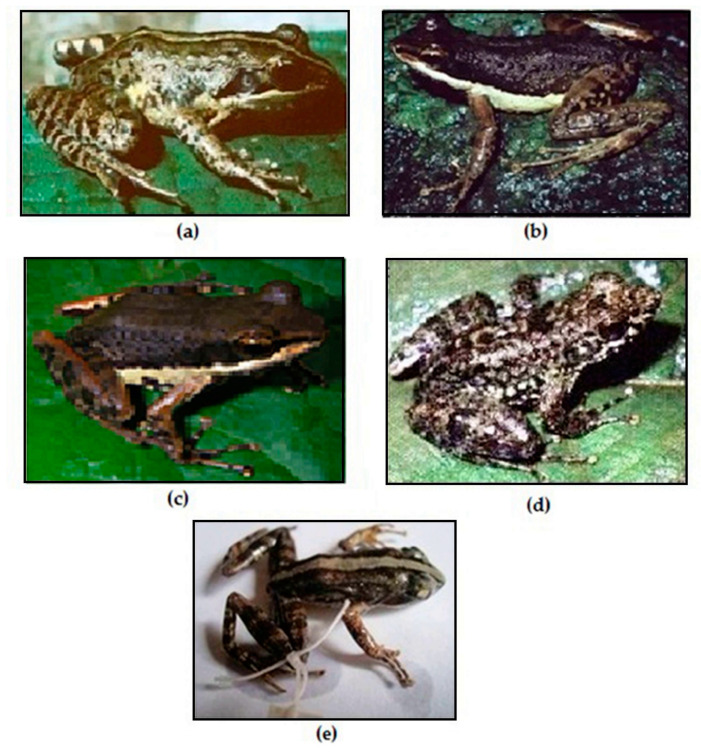
Photos of valid nominal species of the subgenus *Ochthomantis*: (**a**). *Mantidactylus femoralis*, Zahamena (CJR); (**b**). *Mantidactylus ambreensis*, Montagne d’Ambre (CJR), (**c**): *Mantidactylus ambony*, Montagne d’Ambre (CJR); (**d**). *Mantidactylus mocquardi*, Betampona (NR), (**e**). *Mantidactylus zolitschka*, An’ala (NR).

**Figure 3 animals-13-02800-f003:**
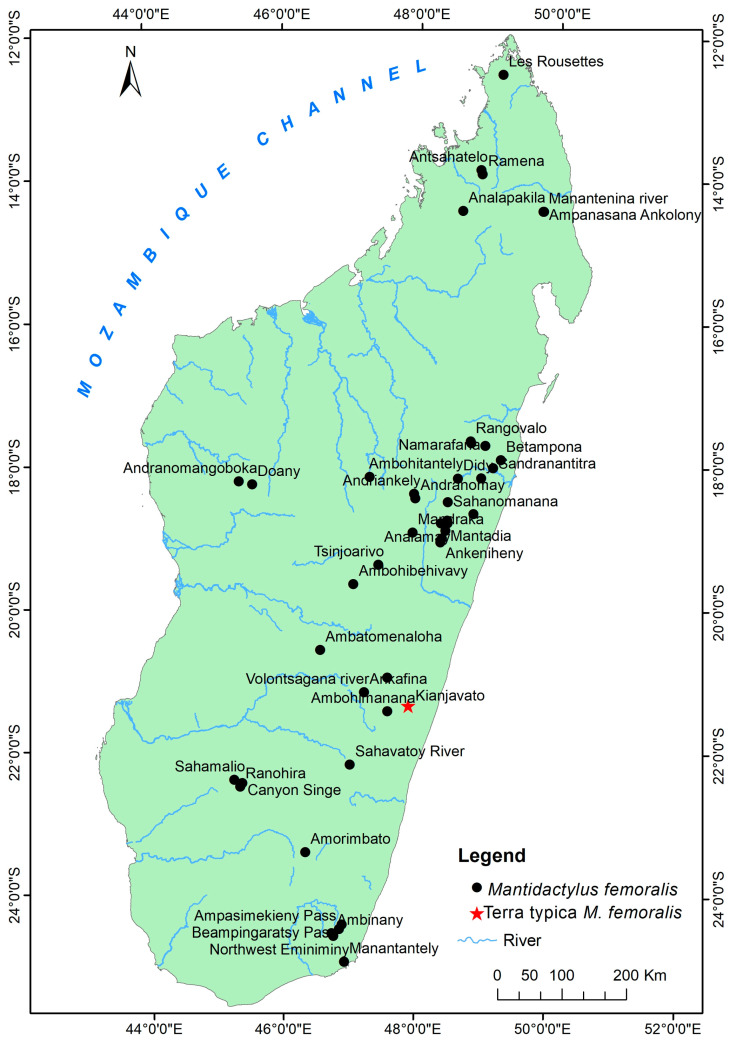
Distribution map of *Mantidactylus femoralis*.

**Figure 4 animals-13-02800-f004:**
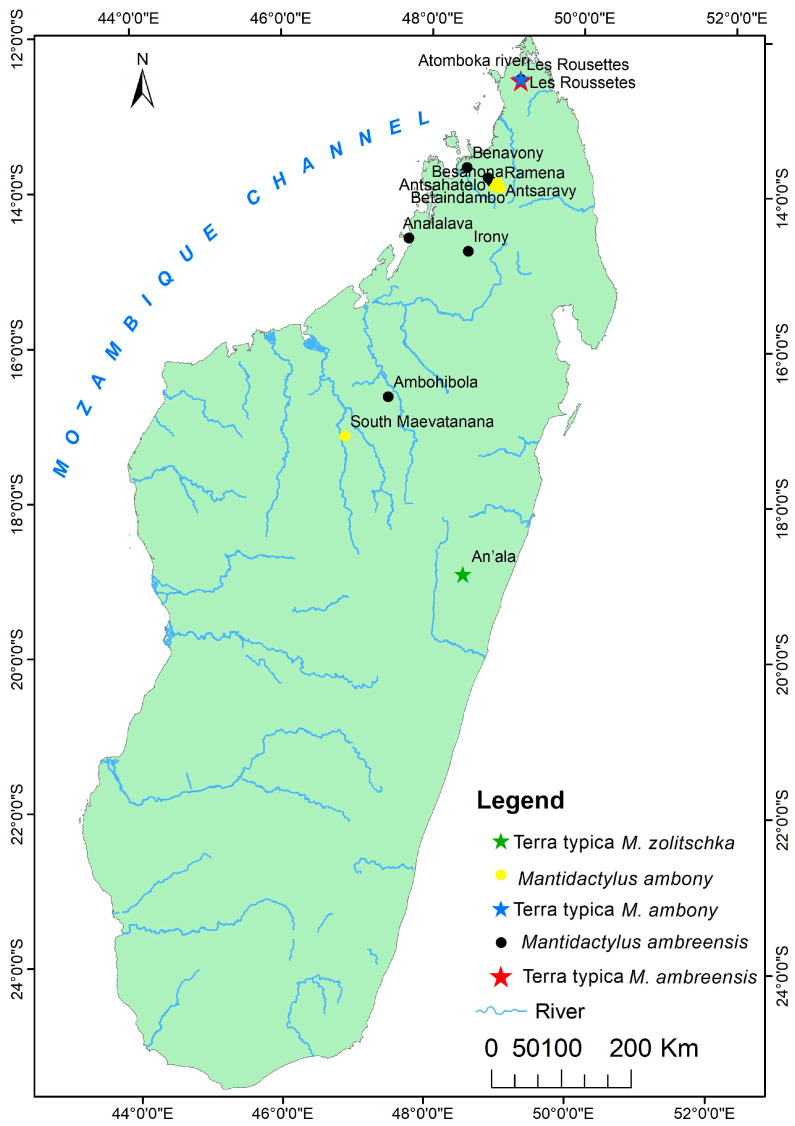
Distribution map of *Mantidactylus ambreensis*, *M. ambony* and *M. zolitschka*.

**Figure 5 animals-13-02800-f005:**
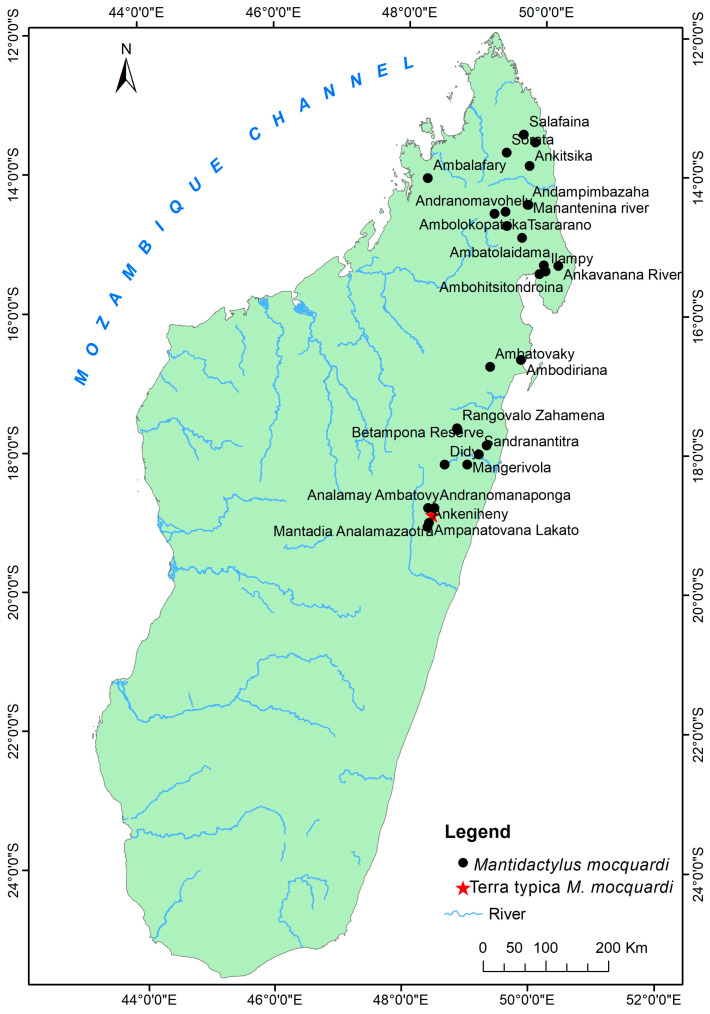
Distribution map of *Mantidactylus mocquardi*.

**Figure 6 animals-13-02800-f006:**
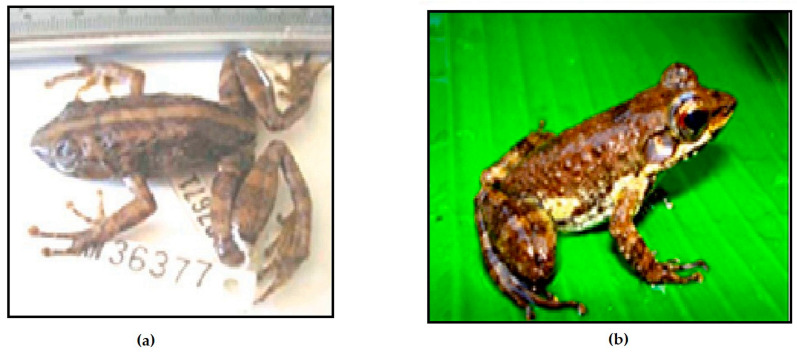
Photos of the resurrected species of the subgenus *Ochthomantis*: (**a**). *Mantidactylus catalai*, Ampasimiekiny Pass (NR), (**b**). *Mantidactylus poissoni*, Mandraka (NR).

**Figure 7 animals-13-02800-f007:**
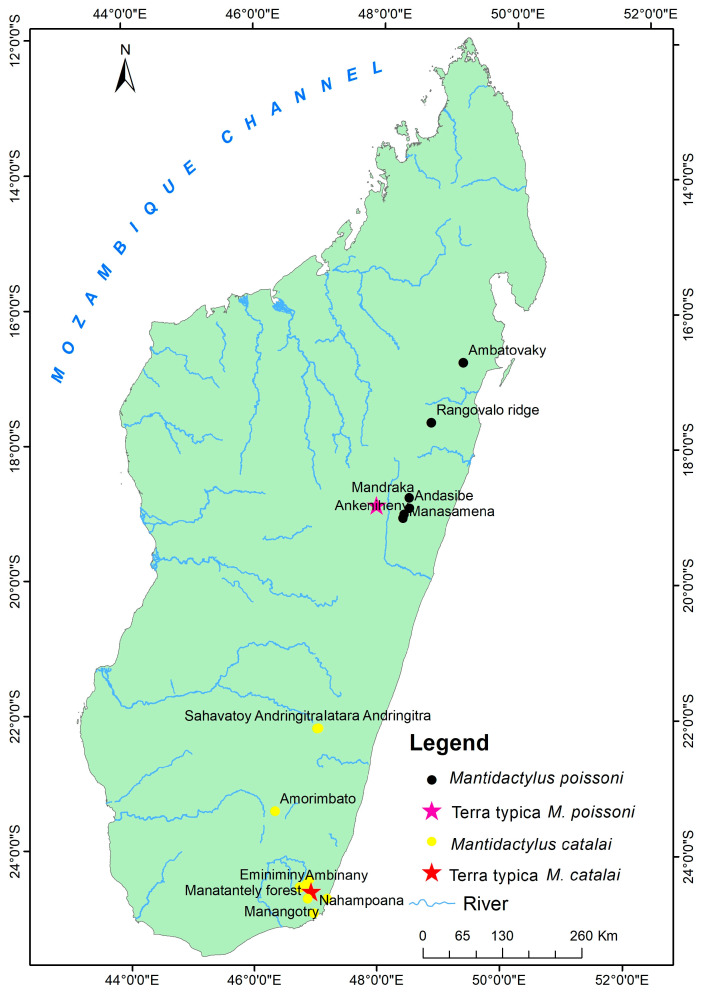
Maps of the distribution of *Mantidactylus catalai* and *M. poissoni*.

**Figure 8 animals-13-02800-f008:**
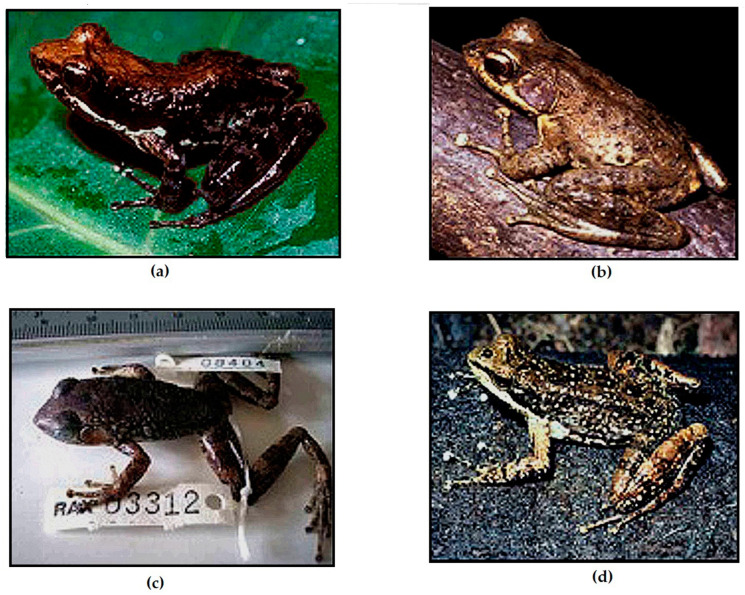
Photos of the new species of the subgenus *Ochthomantis*: (**a**). *Mantidactylus danieli* n. sp. (CJR), (**b**). *Mantidactylus macrotympanum* n. sp. (CJR), (**c**). *Mantidactylus olgae* n sp. (NR), (**d**). *Mantidactylus tavaratra* n. sp. (CJR).

**Figure 9 animals-13-02800-f009:**
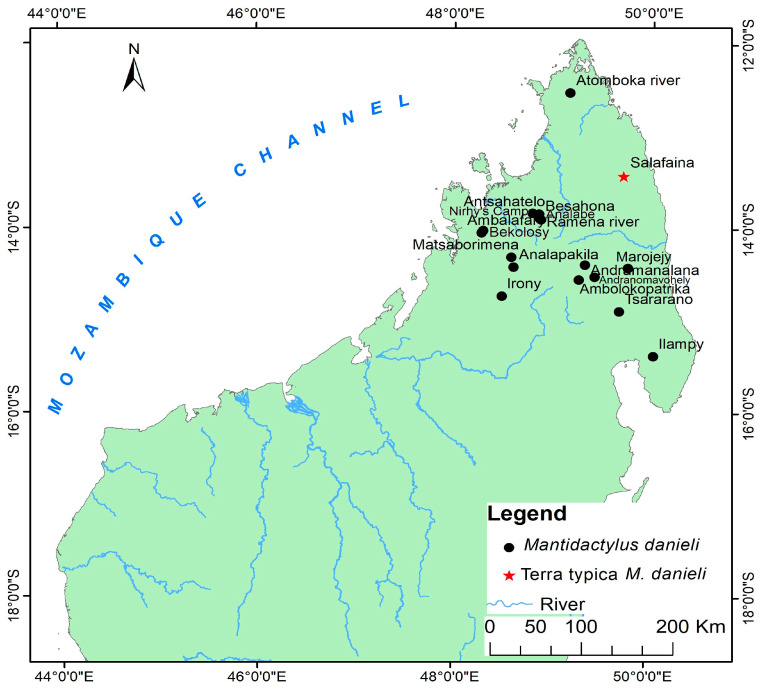
Maps of the distribution of *Mantidactylus danieli* n. sp.

**Figure 10 animals-13-02800-f010:**
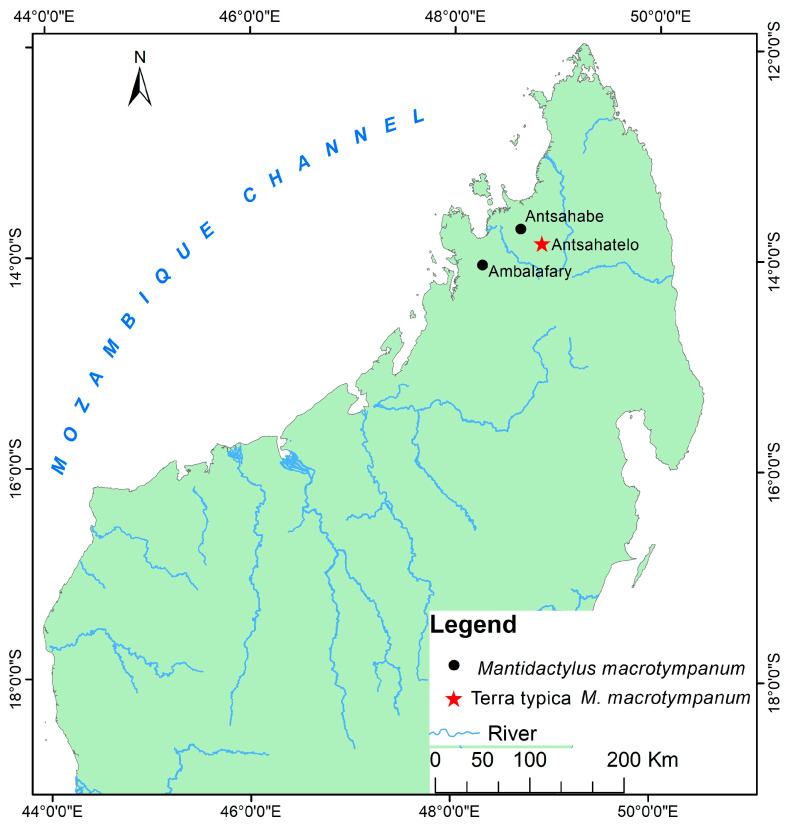
Map of the distribution of *Mantidactylus macrotympanum* n. sp.

**Figure 11 animals-13-02800-f011:**
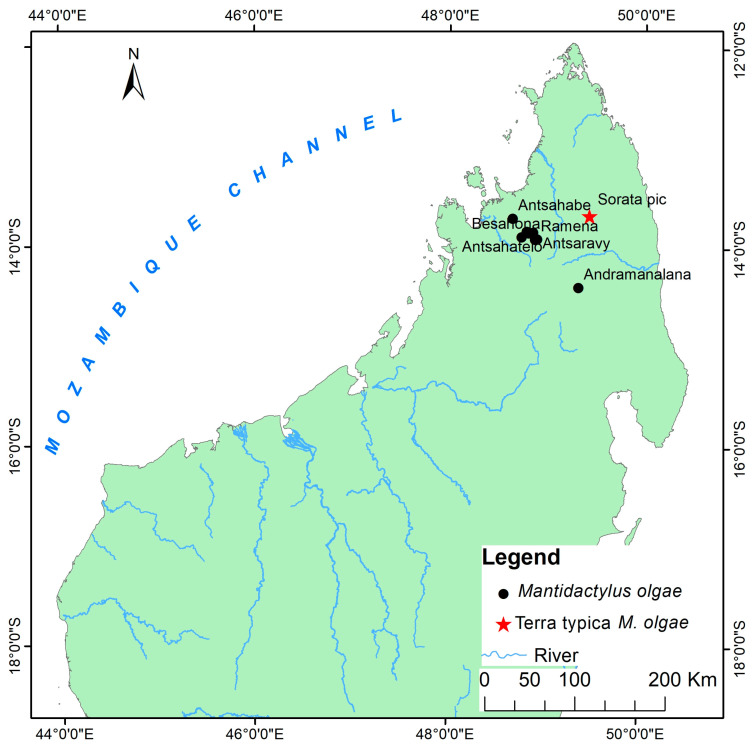
Map of the distribution of *Mantidactylus olgae* n. sp.

**Figure 12 animals-13-02800-f012:**
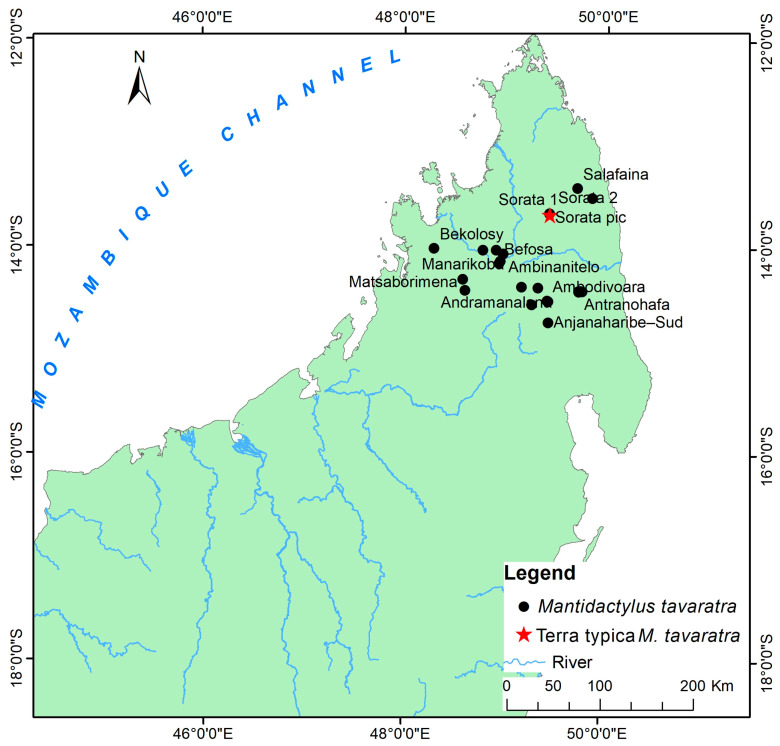
Map of the distribution of *Mantidactylus tavaratra*.

**Figure 13 animals-13-02800-f013:**
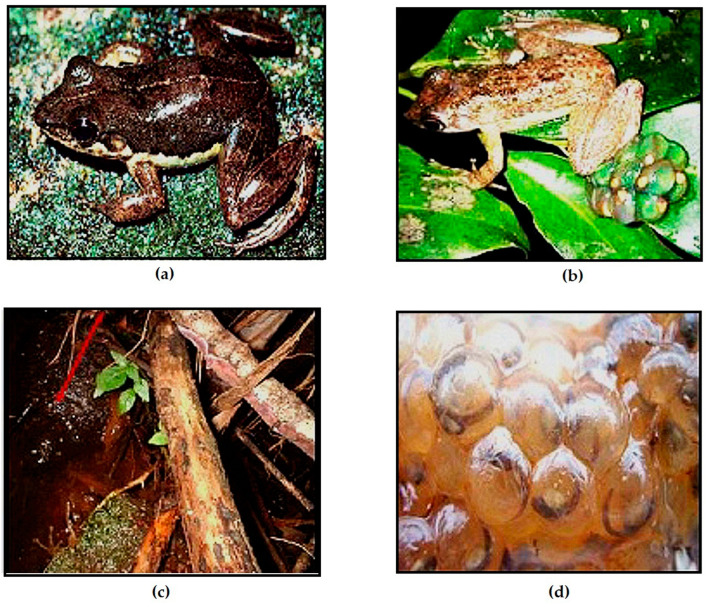
(**a**) Photo of a male *Mantidactylus majori*, Andohahela (CJR), (**b**) egg deposition on leaf overhanging the water and male *M. majori* guarding eggs and practicing parental care, Vatovavy (NR), (**c**) Eggs of *Mantidactylus femoralis* deposited on a falling branch next to riverbank, Manasamena (NR), (**d**) Eggs of *M. ambreensis* laid on a rock, Ambohibola (NR).

**Table 1 animals-13-02800-t001:** Comparison of morphological measurements of the currently valid nominal species of the subgenus *Ochthomantis*. All measurements in mm, F: adult female, M: adult male. For morphological abbreviations see Appendix B.

Size Range	*M. femoralis*		*M. ambreensis*		*M. ambony*		*M. mocquardi*		*M. zolitschka*	
	M	F	M	F	M	F	M	F	M	F
SVL	39.1–62.4	33.0–43.4	38.2–42.2	33.3–39	34.0–37.9	30.0–31.8	46.0–64.3	51.9–63.3	51.9–63.3	26.5–30.6
TD	2.8–6.2	3.0–5.1	2.7–3.7	3.3–5.9	2.2–3.3	4.4–5.0	2.6–5.2	3.5–4.7	3.5–4.7	2.8–3.2
ED	4.4–9.2	3.8–7.5	4.5–5.7	4.2–5.3	4.0–4.9	3.5–4.3	5.7–8.8	6.0–8.2	6.0–8.2	3.3–3.7
EN	2.9–7.0	2.5–4.1	2.8–3.8	2.5–4.0	2.1–3.6	2.3–2.8	3.1–5.6	3.6–5.4	3.6–5.4	1.9–2.7
EST	0.9–3.1	0.4–2.5	1.4–1.9	1.0–2.1	1.3–1.9	1.2–1.7	1.3–3.3	1.6–3.5	1.6–3.5	0.6–1.2
NS	1.0–3.3	1.0–2.8	1.5–2.7	1.1–2.6	1.1–2.0	1.4–2.0	1.2–4.3	1.3–3.8	1.3–3.8	1.3–2.0
NN	3.0–5.5	1.9–4.4	3.2–4.0	2.5–4.5	2.9–3.6	2.6–3.3	3.0–6.6	4.7–5.8	4.7–5.8	1.6–3.3
HW	10.9–20.3	10.7–15.2	11.6–13.8	9.9–12.4	10.6–11.8	9.0–11.0	15.8–23.4	17.1–21.5	17.1–21.5	8.8–10.5
HL	16.2–25.7	14.0–19.3	14.0–16.7	13.8–17.4	13.0–15.5	12.8–14.8	19.7–28.4	22.2–27.5	22.2–27.5	11.2–13.1
HDL	11.5–17.2	10.0–13.4	10.4–13.0	9.0–12.0	9.7–11.7	8.6–9.9	11.4–19.6	13.0–18.1	13.0–18.1	8.0–10.2
RC	8.4–10	5.6–8.7	7.0–8.6	5.5–7.3	6.3–7.8	5.6–6.2	7.7–12.0	9.2–12.1	9.2–12.1	5.2–5.3
FE	20.6–34	15.8–21.6	18.6–20.6	15.4–19.8	18.0–20.2	14.1–16.0	24.2–33.1	27.0–31.9	27.0–31.9	14.3–14.8
TI	21.6–32.7	17.0–23.0	19.8–23.5	17.1–21.3	19.3–20.5	16.0–17.2	24.0–35.4	28.1–32.1	28.1–32.1	14.5–14.7
FT	22–32.1	17.6–23.8	18.4–21.6	16.1–20.6	18.5–20.3	15.3–16.8	23.1–33.7	26.8–32.0	26.8–32.0	19.5–20.9
TA	9.8–14	8.1–11.5	9.7–11.5	8.0–10.6	8.7–10.1	7.6–8.0	9.5–14.6	11.3–14.2	11.3–14.2	6.2–6.8
EO	6.1–10.3	4.6–7.5	6.1–7.5	5.1–7.0	5.6–6.4	4.6–5.7	7.5–11.9	8.8–11.3	8.8–11.3	4.4–4.6
EM	2.2–3.6	1.6–3.2	1.7–2.6	1.7–2.4	1.9–2.4	1.6–1.9	2.2–4.3	2.8–4.1	2.8–4.1	1.9–2.0
EHEAD	5.3–9.5	4.3–8.4	5.4–7.2	4.5–6.4	5.0–5.6	4.4–5.0	6.8–10.8	6.7–9.3	6.7–9.3	4.9–5.4
T1	1.1–2.6	1.0–2.2	1.2–1.8	1.0–1.8	1.1–1.3	0.8–1.4	1.6–2.7	1.8–3.0	1.8–3.0	0.8–0.9
TO3	5.4–9.2	4.5–6.6	4.7–6.0	4.0–5.7	4.8–5.5	3.7–4.6	5.5–8.8	7.1–8.8	7.1–8.8	4.8
TO5	6.0–10	5.3–7.0	5.4–6.6	4.8–6.3	5.2–6.4	4.3–5.4	6.5–10.4	8.0–9.7	8.0–9.7	5.3

**Table 2 animals-13-02800-t002:** Comparison of qualitative morphological characters and biogeography of the currently valid nominal species of the subgenus *Ochthomantis*.

Morphological Characters	*M. femoralis*	*M. ambreensis*	*M. ambony*	*M. mocquardi*	*M. zolitschka*
Snout tip very pointed	No	No	No	Yes	No
Tibiotarsal articulation position	Beyond nostrils	Between eye–nostril	Between eye–nostril	Between eye–nostril	Nostrils
Large tympanum	No	Yes	Yes	No	No
Large digit terminal disc	Yes	Yes	Yes	Yes	Yes
Body with striking granules	No	No	No	Yes	No
Body coloration	Dark brownish	Dark green, brown grayish, brown	Dark green, brown	Black, blackish	Brown
Throat with two parallel marks	Stripe	Bar	Bar	X-like pattern and +-like pattern	Y-like pattern
Dorsum with crossbar	No	No	No	V-like pattern or Y-like pattern	No
Mouth with whitish band	No	Yes	Yes	If present (interrupted)	Yes
Yellowish or whitish shape in groin area	Yes (band oblique)	Yes (band running along flank)	Yes (band running along flanks)	If present (spot)	Yes (narrower oblique band)
Ventral surface coloration	Yellowish	With white spot pigment	With white spot pigment	Blackish, silver pigment	Yellowish
Dorsal surface with black spot	No	No	No	Yes	No
Toe fully webbed	No	No	No	No	No
Elevation (m)	90–1600	200–1150	350–1150	350–1000	840
Distribution	North, Sambirano, east, central highland, southeast	North, Sambirano, northwest	Montagne d’Ambre, Analabe, Maevatanana	Northeast, central east, Betampona, Corridor Ankeniheny/Zahamena, Ambatovy, Mantadia, Anala	An’ala

**Table 3 animals-13-02800-t003:** Morphological measurement diagnostic for the two resurrected species of the subgenus *Ochthomantis*. All measurements in mm, F: adult female, M: adult male. For morphological abbreviations see Appendix B.

Size Range	*M. catalai*		*M. poissoni*	
	M	F	M	F
SVL	51.9–63.3	41.1–45.4	53.4–65.3	30.7–48.2
TD	3.5–4.7	5.0–6.6	3.3–4.7	3.6–6.0
ED	6.0–8.2	5.0–6.2	6.0–7.6	5.8–6.5
EN	3.6–5.4	3.3–4.3	4.1–5.4	3.6–4.2
EST	1.6–3.5	1.8–3.2	1.6–3.3	1.5–2.8
NS	1.3–3.8	1.9–3.3	2.1–4.6	1.8–3.7
NN	4.7–5.8	3.4–4.4	4.6–5.9	4.0–5.0
HW	17.1–21.5	13.9–18.5	16.1–22.2	12.5–17.1
HL	22.2–27.5	17.8–21.1	21.2–27.8	17.7–22.2
HDL	13.0–18.1	12.0–17.9	13.5–20.2	13.0–14.6
RC	9.2–12.1	7.9–12.7	9.4–13.7	9.0–10.0
FE	27.0–31.9	20.6.24.0	29.1–35.8	24.4–24.8
TI	28.1–32.1	20.7–23.6	30.3–37.3	24.3–26.0
FT	26.8–32.0	20.1–24.5	29.1–35.2	22.7–26.6
TA	11.3–14.2	8.4–11.5	12.0–16.1	8.8–11.0
EO	8.8–11.3	6.6–7.7	9.5–11.4	7.4–10.0
EM	2.8–4.1	2.0–3.1	3.1–3.8	2.7–4.0
EHEAD	6.7–9.3	6.3–7.7	7.7–10.2	6.8–9.4
T1	1.8–3.0	1.6–2.3	1.7–3.1	1.8–2.0
TO3	7.1–8.8	5.3–6.4	7.2–11.0	7.3–7.6
TO5	8.0–9.7	5.8–7.2	8.5–11.4	8.3–8.6

**Table 4 animals-13-02800-t004:** Qualitative morphological characters and biogeography diagnostics for two resurrected species of the subgenus *Ochthomantis*.

Morphological Characters	*M. catalai*	*M. poissoni*
Snout tip very pointed	Yes	No
Tibiotarsal articulation position	Between eye–nostril	Nostril–snout tip
Large tympanum	Yes	No
Large digit terminal disc	Yes	Yes
Body with striking granules	Yes	No
Body coloration	Brown	Brown
Throat with two parallel marks	Spot	Single 8-like pattern
Dorsum with crossbar	No	No
Mouth with whitish band	No	No (large pale or multiple white spot)
Yellowish or whitish shape in groin area	No	No
Ventral surface coloration	Clean pattern	Yellowish and whitish pigments
Dorsal surface with black spot	Yes	No
Toe fully webbed	No	No
Elevation (m)	20–1300	600–1450
Distribution	Southeast	Central East, Mandraka

**Table 5 animals-13-02800-t005:** Morphological measurements of the four new species of the subgenus *Ochthomantis* are described herein. All measurements in mm, F: adult female, M: adult male. For morphological abbreviations see Appendix B.

Size Range	*M. danieli*		*M. marcotympanum*		*M. olgae*		*M. tavaratra*	
	M	F	M	F	M	F	M	F
SVL	42.3–53.0	33.9–51.9	–	59.4–62.1	45.8–51.8	33.6–42.8	42.7–63.3	35.7–48.8
TD	3.5–6.5	3.3–5.1	–	7.9–8.3	2.6–4.4	3.4–5.6	2.6–5.0	3.0–5.3
ED	4.9–9.4	4.0–5.9	–	7.9–8.3	4.9–7.4	4.1–6.3	4.7–7.8	4.2–6.0
EN	2.0–4.7	2.5–4.5	–	4.6–5.3	2.7–5.0	2.4–4.2	3.1–5.4	2.7–4.5
EST	0.4–2.6	1.0–2.0	–	2.0–2.4	2.2–3.0	1.7–2.5	0.8–3.1	1.5–2.4
NS	1.1–3.3	1.1–2.5	–	2.5–3.0	1.7–3.3	2.0–2.9	1.9–4.4	1.7–3.2
NN	3.6–5.0	2.8–4.4	–	4.6–5.7	3.0–5.2	2.6–4.9	3.6–6.6	3.3–4.4
HW	15.5–20.4	11.7–17.8	–	20.3–21.7	15.2–17.1	11.3–14.4	15.3–23.3	11.6–15.8
HL	19.0–25.1	15.5–24.4	–	25.8–26.8	18.7–22.3	14.2–20.0	18.1–28.2	16.2–21.6
HDL	11.0–15.5	9.6–15.6	–	17.3–19.8	13.0–15.9	10.1–12.4	10.8–18.4	10.8–15.6
RC	8.1–11.7	6.2–10.4	–	10.7–12.1	7.5–12.4	5.5_7.0	7.5–12.3	6.2–8.1
FE	22.1–26.1	15.8–28.8	–	29.4–34.8	23.6–27.5	16.5–21.4	23.8–34.3	18.7–24.4
TI	23.4–28.1	18.0–31.6	–	31.2–34.3	22.6–26.7	16.0–20.4	23.1–33.6	18.6–24.4
FT	23.5–29.7	19.6–28.1	–	30.5–32.6	22.4–25.8	16.4–20.6	23.1–34.3	19.6–25.1
TA	11.0–14.5	7.2–12.1	–	13.7–14.3	8.8–12.5	7.1–9.6	10.0–14.6	8.2–12.4
EO	7.5–9.5	5.8–9.7	–	10.1–11.7	6.7–8.8	5.6–7.5	10.0–9.9	6.0–8.4
EM	2.5–3.3	1.7–3.3	–	3.4–4.2	2.3–3.4	1.8–3.0	2.5–4.3	1.8–3.5
EHEAD	8.5–8.9	5.3–8.6	–	9.0–10.2	5.8–9.2	4.7–6.2	6.1–10.0	4.5–7.6
T1	1.8–2.2	0.9–1.9	–	1.9–3.0	1.3–2.4	1.0–2.2	1.4–2.9	1.2–2.2
TO3	6.1–7.7	4.3–7.4	–	8.7–9.4	4.7–6.8	3.8–5.1	5.5–9.9	4.5–7.2
TO5	7.0–8.7	4.7–8.5	–	10.1–10.9	5.9–7.8	4.6–6.1	5.8–10.6	5.0–7.8

**Table 6 animals-13-02800-t006:** Qualitative morphological characters and biogeography of the four new species of the subgenus *Ochthomantis* are described herein.

Morphological Characters	*M. danieli*	*M. macrotympanum*	*M. olgae*	*M. tavaratra*
Snout tip very pointed	No	No	Yes	No
Tibiotarsal articulation position	Beyond nostrils	Beyond snout tip	Between eye–nostril	Between eye–nostril
Large tympanum	No	Yes	No	No
Large digit terminal disc	No	Yes	Yes	No
Body with striking granules	Yes	No	Yes (black)	No
Body coloration	Brown with black spots	Brown	Gray	Gray
Throat with two parallel marks	L-like pattern or spot	Bar	Stripe	Stripe
Dorsum with crossbar	No	No	V-like pattern or Y-like pattern	V-like pattern or Y-like pattern
Mouth with whitish band	Yes	Yes	No	Yes
Yellowish or whitish shape in groin area	Yes (band oblique)	White plate-like pattern	No	If present (narrow white band oblique)
Ventral surface coloration	Yellowish	Clean pattern	Clean pattern	With silvery pigments
Dorsal surface with black spot	Yes	No	Yes	Yes
Toe fully webbed	No	Yes	Yes	Slightly
Elevation (m)	350–1580	180–800	600–1700	530–2650
Distribution	North, Sambirano, northeast, northwest	Analabe, Manongarivo	Analabe, Andramanalana	North

## Data Availability

All holotypes are housed at AMNH and paratypes are shared between AMNH and UADBA according to research permits. Specimens used for molecular analyses are also stored in both institutions. Molecular sequences reported here are available in GENBANK.

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
