# Peer review of "Revision of the Subgenus Ochthomantis Frogs from Madagascar (Amphibia: Mantellidae) with the Description of Four Species and Resurrection of Mantidactylus catalai and M. poissoni"

_animals, 2023, doi:10.3390/ani13172800_

Round 1

Reviewer 1 Report

[IMPORTANT: please check the attached Word document with extensive edits that I have directly made in the manuscript, and use it as basis for the revision]

I have completed the review of the paper by Rabibisoa et al. revising the subgenus Ochthomantis.

This is a long awaited paper that was missing for almost 15 years, and will finally allow to make the diversity of Ochthomantis frogs accessible for evolutionary and ecological research, and conservation management. The study is based on a large and adequate amount of data, and the conclusions are largely backed up by the data presented.

The authors have studied morphologically a large number of frog specimens, and identified the majority of them based on these morphological data. Only for some of them are genetic data presented, and bioacoustic analyses are missing. This reliance on a single set of characters (morphology) for identification means that probably, some of the specimens and localities are wrongly assigned to species, but this is normal in comprehensive taxonomic revisions, and there is no doubt that the study represents an enormous leap forward in our understanding of this group of frogs. Importantly, for the four new species described, all holotypes have been genetically characterized, so that the taxonomy is stable and the new species well defined according to state-of-the-art integrative taxonomic approaches.

Unfortunately, the paper in its submitted form contained a series of weaknesses. These largely concerned the writing, which is unfortunate as there are two native speakers among the authors which obviously have not invested sufficient time in cleaning up the manuscript and polishing grammar, style and orthography. Therefore, in order to make this important paper fit for publication and avoid further delays, I went through the submitted Word document and applied all my edits and corrections directly in the text. For some sections, especially the Discussion, this required rewriting many sentences and deleting entire paragraphs of excessively speculative nature (see more on this topic below).

Other than the writing, there are a few important issues with the molecular analysis and with zoological nomenclature, and there is a lack of verifiability because some data (i.e., DNA sequences and alignment) have not been made available. These issues need to be resolved, and a fully revised manuscript including the edits in the attached Word file needs to be submitted for a second round of review which I expect will only require a few additional modifications, especially regarding the interpretation of the molecular results.

Main compulsory revisions:

1. According to Latin grammar, the correct name for Mantidactylus olgai must be olgae. I have made these corrections in the text, but the authors must make sure to apply them also to all figures and supplementary materials, as well as Genbank accessions, wehere this name may appear.

2. Make sure to adhere to the regulations of the Code of Nomenclature: A neotype can only be designated if very clear conditions are fulfilled, and it definitely is not allowed to designate a neotype while the original holotype still exists, as in the case of M. poissoni.

3. Since Animals is an online only journal, it is necessary to register all nomenclatural acts (new species) with Zoobank, and provide the LSID numbers in the manuscript.

4. The Methods say that DNA sequences have been submitted to Genbank. The respective Genbank accession numbers must be presented in the revised manuscript version.

5. Please make the alignment used for phylogenetic analysis available for checking and reanalysis. The phylogenetic tree contains one long-branch clade which might either be an artefact of the analysis itself or a misalignment. This point must be fixed in the second round of review, which will be easy to do, but for this I need to be able to inspect the original alignment.

6. In Supplementary Tables S2-S12, change Gender to Sex.

Minor revisions:

I have made a large number of corrections and edits, and several further comments, directly in the manuscript file. Please download the edited Word file (or request it from the publishers or editor), and use this edited Word file for your revision. Check the tracked changes of which most should be approved (in some cases you may decide to reject them, but then you need to modify the respective sections in another way).

In the Discussion, you will see I have deleted entire paragraphs, where they contained purely speculative considerations that are not backed up by the data and/or analyses of this paper. Keeping these sections would require doing very extensive and complicated additional statistical analyses to prove your points, and this is beyond the scope of this paper. The same by the way is also true regarding the position of Mantidactylus majori. This paper simply does NOT present the data needed to clearly demonstrate its belonging into a different subgenus, so this point needs to be de-emphasized in the Discussion (I have made some suggestions).

I have extensively edited the manuscript, but after accepting (or where appropriate, rejecting) my edits, it is important that the two native English speakers in the team take a careful look especially at the Discussion but also some parts of the results (like description of coloration and natural history) and polish style and grammar.

Reviewer 2 Report

This appears to be the result of a sound study that will make a significant contribution to the knowledge of species diversity within this subgenus. I have only one minor concerns not related to writing.

This manuscript needs to be edited for numerous errors. Moreover, it can be streamlined in places.  Here are some specific suggestions:

1. Have one of the authors who is a native English speaker rewrite the Simple Summary completely. 

2. Streamline the diagnoses so that the writing is consistent. Right now, it switches back and forth between short, descriptive phrases and full sentences. Pick one. If you prefer, you can use this rewritten diagnosis for M. femoralis as a model. It is all short phrases:

Diagnosis: A medium to large–sized Ochthomantis (adult male SVL 31–43.4 mm; adult female 43.0–62.4 mm). Tibio–tarsal articulation reaching beyond nostrils (rarely between eye and nostril); 1.5–2 free phalanges on internal edge of toe 4; width of digit terminal disc ≥ 1.70 disc base; white stripe along superior lip and prominent yellow patch in inguinal region. Distinguished from other subgenus species by following combination of characters: M. ambreensis by yellow line or patch in inguinal region; M. poissoni by absence under eye of large, white spot or partly fused white spots on upper lip; M. mocquardi, M. catalai, M. olgai, and M. tavaratra by presence of 1.5–2 free phalanges on internal edge of toe 4, yellow patch in inguinal region, and tibio–tarsal articulation reaching beyond eye; M. zolitschka by its large size (SVL ≥ 33 mm) and yellow patch in inguinal region; M. danieli by yellow patch in inguinal region and width of the digit terminal disc ≥ 1.70 disc base; M. macrotympanum by smaller adult male SVL (< 60 mm). Character diagnostics summarized in Tables 1, 2.

3. Cut back on the use of commas. Currently, they are rampant and often incorrectly applied. As a general rule, use commas only when separating things of equal value (e.g., items in a list; clauses in a compound sentence, each with its own noun and verb) or to distinguish a subordinate clause that represents a sudden change in thought (e.g., "I ran, although it hurt to do so"). An example of an error repeated throughout the manuscript is in the Abstract (line 5) after "field." No subject follows "and" in this case and thus is not a compound sentence.

4. Make sure all sentences make sense. For example, on p. 2, 2nd full paragraph, line 9, this sentence is poorly written and confusing:

“Reviewing mantellid species using morphological character diagnostics are already demonstrated in influencing speciation by [12], this is why the utility of studying cryptic species using both diagnostic morphological characters and molecular data is important to solve the classification problematic of the subgenus Ochthomantis.”

Suggested revision: “A past review of mantellid species using morphological characters has demonstrated that undescribed species likely exist [12]. Therefore, a study utilizing both morphological characters and molecular data is important to resolve the classification of the subgenus Ochthomantis.”

5. "Dorsal" is an adjective, not a noun. For example, in Table 2, change "Dorsal with crossbar" to "Dorsum with crossbar." Also, phrases such as "dorsal body," while informative, are not technically correct because there is only one body. It should be "dorsal surface of body" or some such thing.

6. Check carefully for misspellings. Some examples include "length" (p. 10, line 18 and other places), "semi-aquatic”( p.17, paragraph 3, line 1 and other places), "summarized" (p. 8, last line and other places), "burn-in" (p. 3, 2nd full paragraph, line 16), and "rainforest" (p. 11, last paragraph, line 1). Similarly, make sure real numbers have decimals rather than commas (e.g., p. 13, last paragraphs, lines 1-2).

7. Make sure that any word chosen is clear and precise in its meaning. For example, on p. 19, 2nd full paragraph, line 9, what is meant by “stains?” Does this mean patches of color? "Stains" implies a foreign substance has been spilled on it and left a stain. On the same page, line 10, does “clear” means transparent or simply devoid of markings? There are numerous other cases such as these that need to be carefully edited.

8. While a reader can figure out the meaning of some incorrect phrases, these phrases  still need to be fixed for clarity and precision. For example, p. 2, 2nd full paragraph, line 6 “candidates for unrecognized additional species diversity” is incorrect. The groups are candidates for "species," not for "species diversity."

9. Phrases like "northern of Madagascar" (M & M, line 6) are incorrect. Say either "northern parts of Madagascar" or "northern Madagascar."

10. Some additional specifics:

a. p. 3, line 6. Why is the reference not sequentially numbered like the others?

b. In Table 2 on p. 10, I do not understand how a frog (M. mocquardi) can have a slightly fully webbed toe. "Slightly" and "fully" are contradictory.

c. p. 14, 2nd paragraph, line 4 and other places. Why give both reciprocals of the head width/length ratio? It seems redundant.

d. p. 19, line 26. Replace "femur" with "thigh" or "femoral region." The femur is technically the bone.

e. line 16. What is meant by “except the holotype?” Was that the only specimen missing the character?

Round 2

Reviewer 1 Report

I am pleased with the revision by the authors. In particular, I am happy that the anomaly (placement of macrotympanum) in the phylogenetic tree has been removed, and that the authors are making all of the genetic and morphometric data available for re-use and re-analyses by subsequent students and researchers who may wish to work on this genus.

Before final acceptance, it is obligatory to add Genbank accession numbers, but as the authors write, these have already been submitted and so they can be expected to become available soon.

I have only a few minor corrections, the first of which is however important:

Line 1265: "Etymology: The specific name macrotympanum is a Latin adjective referring to the very big tympanum of this species, compared to other."

--> This is wrong, macrotympanum is not an adjective, as "tympanum" is a Latin noun. Please  correct this! The sentence could instead read:
"Etymology: The specific name macrotympanum is a composed of the Latin words macro (large) and tympanum (tympanum), referring to the very big tympanum of this species. The name is used as noun in apposition.

Line 1150 regionclean pattern --> a space is missing

Line 1186: is a patronym

Line 1247: 57.90 and 62.10  --> change to “57.9 and 62.1”

Line 1620 We were noticed  --- We noticed
